

# Photochemical method for removing methane interference for improved gas analysis

Merve Polat[1], Jesper Baldtzer Liisberg[2], Morten Krogsbøll[1], Thomas Blunier[2], and Matthew S. Johnson[1]

[1]Copenhagen Center for Atmospheric Research, Department of Chemistry, University of Copenhagen, Universitetsparken 5, DK-2100 Copenhagen Ø, Denmark
[2]Physics of Ice Climate and Earth, Niels Bohr Institute, University of Copenhagen, DK-2100 Copenhagen Ø, Denmark

**Correspondence:** Matthew S. Johnson (msj@chem.ku.dk)

**Abstract.** The development of laser spectroscopy has made it possible to measure minute changes in the concentrations of trace gases and their isotopic analogs. These single or even multiply substituted species occur at ratios from percent to sub-ppm and contain important information concerning trace gas sources and transformations. Due to their low abundance minimizing spectral interference from other gases in a mixture is essential. Options including traps and membranes are available to remove many specific impurities. Methods for removing $CH_4$, however, are extremely limited as methane has low reactivity and adsorbs poorly to most materials. Here we demonstrate a novel method for $CH_4$ removal via chlorine-initiated oxidation. Our motivation in developing the technique was to overcome methane interference in measurements of $N_2O$ isotopic analogs when using a cavity ring-down spectrometer. We describe the design and validation of a proof-of-concept device and a kinetic model to predict the dependence of the methane removal efficiency on methane concentration $[CH_4]$, chlorine photolysis rate $J_{Cl_2}$, chlorine concentration $[Cl_2]$, and residence time $t_R$. The model was validated by comparison to experimental data and then used to predict the possible formation of troublesome side- and by-products including $CCl_4$ and HCl. The removal of methane could be maintained with a peak removal efficiency > 98 % for ambient levels of methane at a flow rate of 7.5 ml min$^{-1}$ with $[Cl_2]$ at 50 ppm. These tests show that our method is a viable option for continuous methane scrubbing. Additional measures may be needed to avoid complications due to the introduction of $Cl_2$ and formation of HCl. Note that the method will also oxidize most other common volatile organic compounds. The system was tested in combination with a cavity ring-down methane spectrometer, and the developed method was shown to be successful at removing methane interference.

## 1 Introduction

Infrared absorption is a fast, convenient, and non-destructive approach for measuring gas composition used in a wide range of applications. High-resolution instruments based on specific rovibrational transitions are becoming available to characterize the abundance of rare isotopocules within gases. Laser spectroscopy has entered territory that has been the exclusive domain of mass spectrometry. While recent advances in the field can give the impression that new laser-based instruments can be used in a "plug and play" manner, there are still limitations to the accuracy and reproducibility of the measurements.

In a recent study investigating the performance of currently available laser spectroscopic $N_2O$ isotope analyzers (Harris et al. (2020)), a number of interferences from other trace gases were identified, arising from spectral overlap of $N_2O$ and the rovi-


brational spectra of the other gases. The consequence was an offset in the measured isotopocule abundance value arising exclusively from ambient levels of methane for a Picarro G5131-i cavity ringdown-based instrument that determines $\delta^{15}N$, $\delta^{15}N\alpha$, $\delta^{15}N\beta$, and $\delta^{18}O$ for $N_2O$. These instruments are often used to measure biological activity in agricultural soils, (Ibraim et al. (2019a), Wolf et al. (2015)), which can help establish a isotopic signature for different $N_2O$ pathways.

Changes in isotopic compositions are commonly accompanied by changes in the abundance of the interfering gases such as
$CH_4$, $CO_2$, and water vapor (Erler et al. (2019), Ibraim et al. (2019b)). These variations complicate measurements. In some samples, changes in $CH_4$ and $CO_2$ can exceed 1.8 ppm and 200 ppm, respectively (M. Zimnoch and Rozanski (2010)). For the instrument described in Harris et al. (2020), these variations will result in an error in measuring $N_2O$ isotopologue abundances of more than 5 ‰ (Harris et al. (2020)). It is desirable to increase the accuracy of the results by controlling these interferences. One solution is multi-line analysis, or careful measurement of the interfering gas(es) with a second instrument, along with a
careful determination of the calibration curve.(Kantnerová et al. (2020)) These options are not desirable for all applications as they either require a redesign of the instrument or investment in additional equipment. A more direct and practical method would be to remove the interfering species from the sample.

For continued measurements, well-established methods including chemical traps and membranes are readily available for the removal of $CO_2$, CO, and humidity. However, to the best of our knowledge, no method for continuous removal of methane
is available with the exception of catalyzed combustion (Cullis and Willatt (1983)) which requires high temperatures and the addition of oxygen thereby altering the gas matrix. It was desired to develop a method for removing $CH_4$ and potentially other VOCs in a manner that would only introduce minimal changes to the matrix composition.

Inspiration for the method investigated in this work was taken from the oxidation pathways taking place in the atmosphere (Pugliese (2018)). The majority of methane is oxidized through an initial reaction with OH radicals (Rigby et al. (2017)) that
results in the formation of $H_2O$ and $CH_3$ radicals. However, the chlorine radical is a potentially important agent in initiating chain reactions: Generally, the reaction rates of Cl with VOCs exceed the analogous ones with OH by at least one order of magnitude. The rate constant for methane's reaction with Cl radicals is $1.07 \cdot 10^{-13}$ cm$^3$ molecules$^{-1}$ s$^{-1}$ (Bryukov et al. (2002)) and with hydroxyl radicals is $6.20 \cdot 10^{-15}$ cm$^3$ molecules$^{-1}$ s$^{-1}$ (Bonard et al. (2002)). The reason for the limited role of chlorine in the global atmosphere is that it's concentration on average is three or four orders of magnitude lower than OH,
although it can have an impact in the stratosphere and in marine and polar environments.

The mechanism for Cl-initiated methane oxidation technology proposed in this study is outlined in reactions: (R1) -(R6).

$$Cl_2 + h\nu \rightarrow 2Cl \tag{R1}$$

$$Cl + CH_4 \rightarrow CH_3 + HCl \tag{R2}$$

$$CH_3 + O_2 + M \rightarrow CH_3O_2 + M \tag{R3}$$

$$CH_3O_2 + Cl \rightarrow CH_3O + ClO \tag{R4}$$

$$CH_3O + Cl \rightarrow HCHO + HCl \tag{R5}$$

$$HCHO + Cl + O_2 \rightarrow CO + HCl + HO_2 \tag{R6}$$



**Table 1.** Table summarizing experiments and setups.

| Setup | Experiment |
|---|---|
| HPXL | A |
| STH-MFC-PD | B |
| STH-PD | C, D, E |
| MTH-PD | F, G, H, I |

**Table 2.** Table summarizing experimental conditions.

| Flask name | $CH_4$ | $Cl_2$ | $N_2O$ | Matrix composition | Flow range |
|---|---|---|---|---|---|
| | / ppm | / ppm | / ppb | | / (ml min$^{-1}$) |
| A | 0 | $100 \pm 2.5$ | 0 | >99 % $N_2$ | 6-23 |
| B | $2.003 \pm 5 \cdot 10^{-4}$ | 0 | 323 | Atmospheric air | 1-29 |
| C | $78 \pm 2$ | 0 | 0 | 20.95 % $O_2$ + >79 % $N_2$ | 0.3-1.2 |
| D | 0 | 0 | 509 | 0.95 % Ar + 20.95 % $O_2$ + >78 % | 28-50 |

   We demonstrate a novel method for $CH_4$ removal through chlorine initiated oxidation. Using four experimental setups, we show that methane removal is highly dependent on the flow, chlorine mixing ratio, and light source. We developed a simple

kinetic model to predict the removal efficiency as a function of the four key parameters in the system; $[CH_4]$, $J_{Cl_2}$, $[Cl_2]$, and residence time $t_R$. The model includes essential reactions and additional estimated radical-wall reactions. Two approaches for estimating the photodissociation rate of $Cl_2$ are presented. The goal is to determine the effect of these variables and achieve the desired methane removal efficiencies by optimizing the parameters. The goal is to achieve removals above 99 % for methane at low to ambient concentrations. With the method developed and refined, a final set of experiments is conducted applying

the method to measurements of $N_2O$. The measured isotopic data of the isotopes subject to methane interference, $\delta^{15}N\alpha$, and $\delta^{18}O$, are compared to the stable values derived from correcting according to the measured methane levels.

## 2   Method

### 2.1   Experimental approach

#### 2.1.1   Methane experiments

Four different variations of the setup are used; High-Pressure Xenon Lamp (HPXL) setup (Figure B1a), Single Tube Hexagonal Photochemical Device setup with MFC (STH-PD-MFC) setup (Figure B1c) and without Flow-Controlled Chlorine Waste (STH-PD) setup (Figure B1b) and the Multiple Tubes Hexagonal Photochemical Device (MTH-PD) setup (Figure 1). Table 1 summarizes the setups and the experiments.





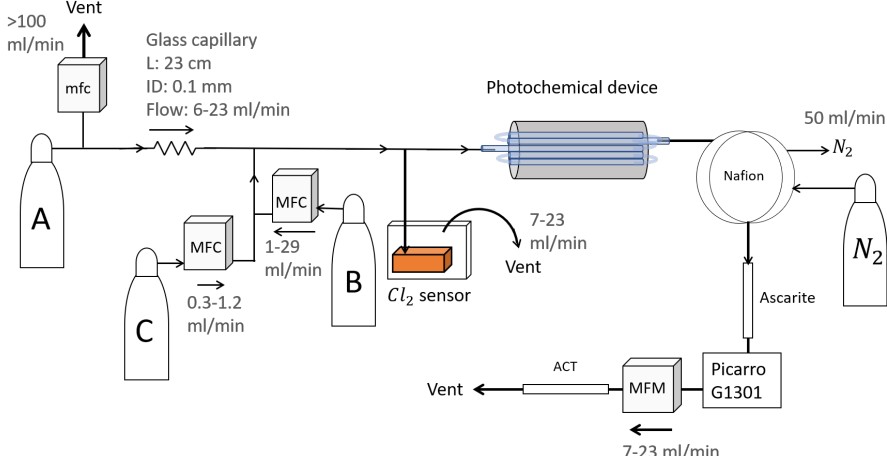

**Figure 1.** MTH-PD setup. The setup is used for experiments F, G, H, and I. ACT: Activated Carbon Trap. MFM: Mass Flow Meter. MFC: MKS Mass Flow Controller GE50A. mfc: manual flow controller. Table 2: Gas flask

[$Cl_2$] is supplied from an external tank labelled flask A (see Table 2 for gas flask). Atmospheric air, flask B, is combined
with an enriched source of [$CH_4$] in flask C to generate various levels of [$CH_4$]. A chlorine sensor is placed outside the main
flow line to reduce the volume of the setup and allow for increased time-resolution.

**Manifold**

The system, Figure 1, has a manifold combining flows from two channels: the sample channel and the chlorine gas channel.
The sample channel provides a flow of methane in air by combining flow from flasks B and C, see Table 2. Variable levels
of [$CH_4$] can be maintained by controlling the ratio between flows. The flow containing methane and chlorine gas is split at a
T-piece, where the main flow proceeds through the photochemical device with excess gas going past a $Cl_2$ sensor. (Chlorine
Gas Detector 0-20 ppm $Cl_2$).

**The Photochemical device**

The photochamber for the HPXL setup uses a quartz tube 20 cm in length and 12.7 mm in outer diameter. The STH-PD,
STH-PD-MFC and MTH-PD setups use a photolysis chamber consisting of 420 LED diodes with peak emission at 365 nm
(Figure 1, B1b and B1c) with the circuit board mounted together in a hexagonal cylinder (illustrated in Figure B2)

The 420 LED diodes are connected in parallel. At the maximum voltage of 3.8 V each consumes 13.2 mA resulting in a
total power of 21 W.

In setup MTH, Figure 1, the $t_R$ in the chamber is increased by a factor of 2.7 by substituting a single quartz tube with seven
smaller quartz tubes in hexagonal shape for optimal packing comprising five tubes OD: 8.33 mm, ID: 6.33 mm and L: 20 cm
and two tubes with dimensions OD: 8.00 mm, ID: 6.00 mm and L: 25 cm.


The tubes were connected in series via Tygon tubes, Tygon R3603, of length 5 cm. The insides of these tubes were coated with krytox, to prevent reaction with $Cl_2$.

### 2.1.2 Post photolysis scrubbing

After the photochemical device the sample passes through a 35 cm Nafion membrane (TT-030 from Perma Pure LLC). The dried sample then passes through an ascarite trap consisting of a central layer of NaOH between two layers of $MgClO_4$ separated by glass wool. These types of traps are normally used for the removal of $CO_2$ and $H_2O$ (Harris et al. (2020)), but they were found to likewise remove HCl and $Cl_2$. The gas stream then flows into a cavity ringdown spectrometer (CRDS), the Picarro model G1301. A nominal flow of 15 ml min$^{-1}$ was maintained with the exception of experiments involving variation

in $t_R$ when this flow was changed accordingly. At the outlet of the Picarro G1301 an activated carbon trap labeled "ACT" is attached, which is mainly used for scrubbing chlorinated species (Ryu and Choi (2004), Milchert et al. (2000)).

### 2.1.3 $N_2O$ Experiments

A final set of experiments is conducted using a Picarro CRDS model G5131-i, capable of measuring $N_2O$ mixing ratio and isotopic abundance. These experiments were performed to validate the effect of the removal of $CH_4$ on the measurement of

105 $N_2O$. These experiments were done in two sets using the setups Figure B1e and Figure B1f. The difference between the two setups was the inclusion of a sofnocat trap in Figure B1f. The sofnocat trap is used to oxidize the CO product (Harris et al. (2020)) and was prepared with 1.25g of sofnocat contained in a 1/4" SS tube of length 8 cm kept in place by glass wool.

## 2.2 Theoretical approach

### 2.2.1 Kintecus, Version 6.8

A model is made with the program Kintecus, version 6.8, (Ianni (2012)) to investigate the reaction mechanisms in the photochemical device. The model contained the relevant reactions with rates for chlorine atom production and removal, methane oxidation, and formation of chlorinated species. The model was kept as simple as possible while still including the relevant reactions. The reactions used in the model are found in Tables E1 - E3. A simplified reaction-scheme is shown in Figure 2. A continuous flow was simulated by setting the initial and external concentrations of gases flowing through the chamber to the

same value. This is done for the gases: $Cl_2$, $CH_4$, $N_2$ and $O_2$. A copy of the model parameters is available in the Appendix C.



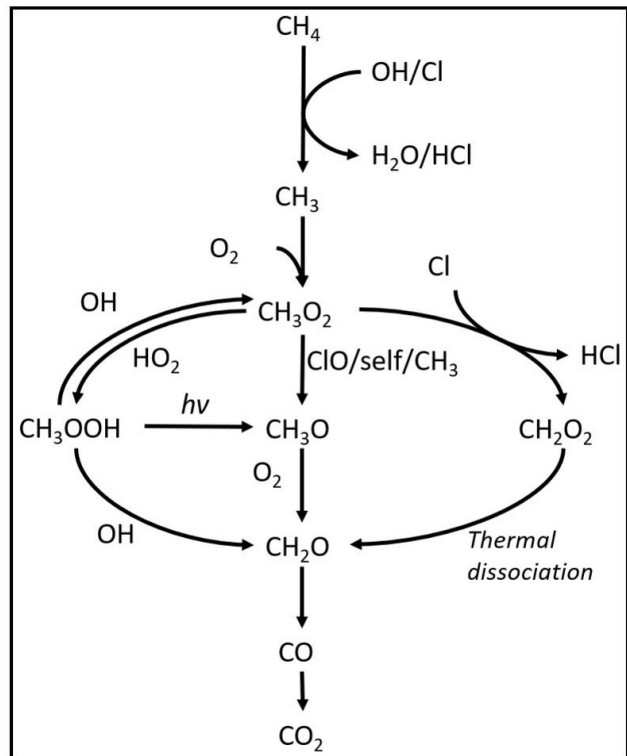

**Figure 2.** Reaction scheme for the oxidation of methane to $CO_2$. $CH_3O_2$ self reactions lead to the formation of $CH_3O$.

**Radical wall reactions**

A set of radical terminating reactions is incorporated in the model to account for reactions on the walls of the quartz tube.

$$Cl \rightarrow \frac{1}{2}Cl_2 \qquad \text{(R7)}$$

$$ClO \rightarrow \frac{1}{2}Cl_2 + \frac{1}{2}O_2 \qquad \text{(R8)}$$

$$OH \rightarrow \frac{1}{2}H_2O + \frac{1}{4}O_2 \qquad \text{(R9)}$$

$$HO_2 \rightarrow \frac{1}{2}H_2O + \frac{3}{4}O_2 \qquad \text{(R10)}$$

The wall reactions are assumed to be diffusion limited. The diffusion length is calculated as the average distance from the wall. The diffusion length and rate were calculated using equations (1) and (2), respectively. The estimate of the diffusion rate is described in detail in the section C1. The diffusion constants, diffusion lengths, and estimated wall reaction rates are shown in Table C1.

$$l = r \cdot \left(1 - \frac{1}{\sqrt{2}}\right) \qquad \text{(1)}$$





Where $l$ in the diffusion length and r being the inner radius of the tube.

$$k = \frac{4 \cdot D}{l^2} \tag{2}$$

Where $D$ is the diffusion constant (see table C1).

## Model results

The outputs from the model are the photodissociation-rate, $J_{Cl_2}$, and the abundance of [Cl] and the production of $CCl_4$ as an indicator of the production of unwanted side-products.

### $J_{Cl_2}$ estimation

The chlorine photolysis rate, $J_{Cl_2}$, is estimated in two ways, which is described in more detail in the section C2. The first approach is to fit $J_{Cl_2}$ to reproduce the observed removal efficiencies from the experimental results. These fits were performed for experiments investigating the effect of power.

A second approach is to estimate $J_{Cl_2}$ by relating it to the electric power going through the circuit, $P_{IN}$. Based on our observation, a second-order polynomial provided the best fit to describe the effective light output, $P_{eff}$ as a function of $P_{IN}$.

$$P_{eff}(P_{IN}) = (a \cdot P_{IN} + b) \cdot P_{IN} \tag{3}$$

Where the constants $a$ ($W^{-1}$) and $b$ (unitless) are experiment-dependent constants that scale the effective light output $P_{eff}$ in W. From the effective power output, the photolysis rate $J_{Cl_2}$ is calculated by eq.(4).

$$J_{Cl_2}(W) = P_{eff}(P_{IN}) \cdot J_{scale} \tag{4}$$

$J_{scale}$ ($J^{-1}$) is the scaling-factor and was calculated from the cross-section of $Cl_2$, the wavelength distribution of the generated light, and the expected photon density. The density of photons depends on the volume and cross-section of the tube within the photochemical device. $J_{Cl_2}$ is fit to the data collected for some of the experimental steps for exp. D and I. Exp. D reflects the Single Tube system ( STH-PD and STH-MFC-PD setups) while experiment I reflects the optimized Multiple Tubes system ( MTH-PD setup). From the fitted $a$, $b$, and calculated $J_{scale}$ the photolysis rate could be calculated for the other experiments.

## 3 Results and Discussion

### 3.1 Experimental results

The findings are based on twelve experiments, named A-L, containing multiple steps of turning on the photolysis under different conditions. These steps will be referred to by their experimental letter and their number, eg. experiment 3 step 5 would be exp. C5. An overview of the settings and resulting removal efficiencies can be seen in Appendix Tables D1 - D3. Table 1 gives an overview of the experiments. Experiment H, Figure 3, was carried out with constant $[CH_4]_{initial}$, and $[Cl_2]$ at $2.000 \pm 0.003$





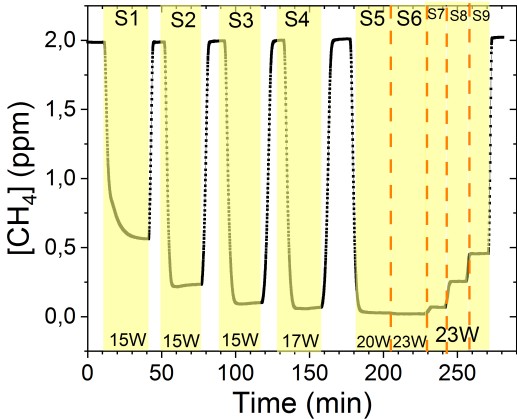

**Figure 3.** Exp. H. The [CH$_4$] is seen as a function of time. The highlight indicates the illumination times. In addition, the experimental step is indicated at the top and $P_{IN}$ (W) is indicated at the bottom

ppm and 50.5 ppm. In addition, the settings for $t_R$ and $P_{IN}$ were varied. As seen in Figure 3 for exp. H1-H4, the efficiency

of the system is improved as the $P_{IN}$ is increased. Starting with H5 a fan is installed to limit temperature. $P_{IN}$ was kept at the same level while the residence time in the chamber was decreased. The three steps, (H1-H3), were carried out with constant $P_{IN}$ at 14.8 W with $t_R$ ranging from 164-350 s. $t_R$ was kept at 350 s for experiments H3-H6. Furthermore, $P_{IN}$ was varied within the range of 14.8-22.8 W.

Two issues affected the results. First, the system was not initially stable. We believe this is due to a build-up of moisture on

the glass-walls, coming to equilibrium after the first step. Seocnd, there is a small continuous pressure drop in the Cl$_2$ bottle, which leads to a decrease in Cl$_2$ and an increase in CH$_4$. We have accounted for this, but it contributes to the uncertainty of our reported Cl$_2$.

**Effect of Residence Time ($t_R$ (s))**

Increasing the residence time results in increased removal of methane, shown in Figure 4a. The $t_R$ was investigated in the

165 Single and Multiple Tube systems. The same flow rate yields a longer $t_R$ for the multiple tube setup due to the 2.7 fold volume increase. The expected trend of asymptotically approaching 100 % can be seen for exp. H, where the high $P_{IN}$ approaches more quickly. The effective light output and $t_R$ are lower for experiments B, C and D compared to H. The resulting removal of methane is accordingly lower. Increasing the $t_R$ is an easy way of enhancing the removal but at the expense of a slower response time of the system.





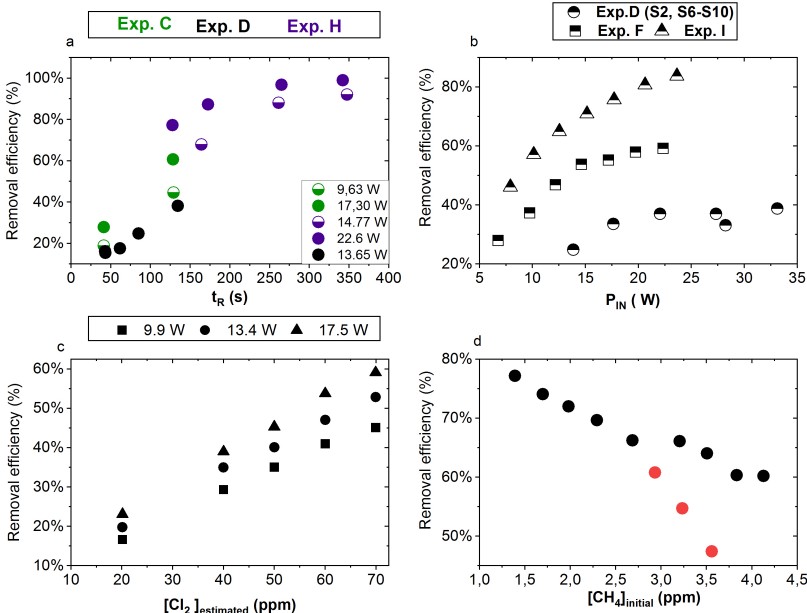

**Figure 4. a.** RE% of methane plotted against $t_R$ (s). The result originates from the three experiments, C (Green), D (Black) and H (Violet). The experiments have different settings in $P_{IN}$, [CH$_4$] and [Cl$_2$]. **b.** RE% of methane plotted against $P_{IN}$ (W). The result are from the three experiments, D (Circle), F (Square) and I (Triangle), which have different settings in $t_R$, [CH$_4$] and [Cl$_2$]. **c.** The Figure presents the methane RE% as a function of the chlorine mixing ratio. Step one, at 30 ppm [Cl$_2$], is an example of start-up-deviation, therefore, it is removed. The points represent the three different $P_{IN}$ of the photochemical device. **d.** The RE% of methane is displayed as a function of the initial methane concentration with the remaining fixed parameters such as [Cl$_2$] mixing ratio, $t_R$, and $P_{IN}$ input. The three red points in the Figure represent steps suffering from start-up-deviation.

## Effect of Power input ($P_{IN}$ (w))

The results from experiments with power variations are shown in Figure 4b . As presented for exp. D the system reaches a maximum removal efficiency, such that increasing the power does not yield significantly higher removal efficiencies. This effect also appears to occur during exp.F. Comparing exp. F to D it is evident a higher removal efficiency has been reached thanks to the longer $t_R$ and higher [Cl$_2$].

## Effect of [Cl$_2$]

Exp. E determined the effect of changing [Cl$_2$], see Figure 4c . [Cl$_2$] is set between 20 ppm and 70 ppm. Higher [Cl$_2$] levels result in an increased methane removal rate. The resulting removal efficiency is still below 60 % and the RE% appears to be linear with [Cl$_2$]. Given the result from exp. E the level of [Cl$_2$] was set to 50 ppm for the remaining experiments.


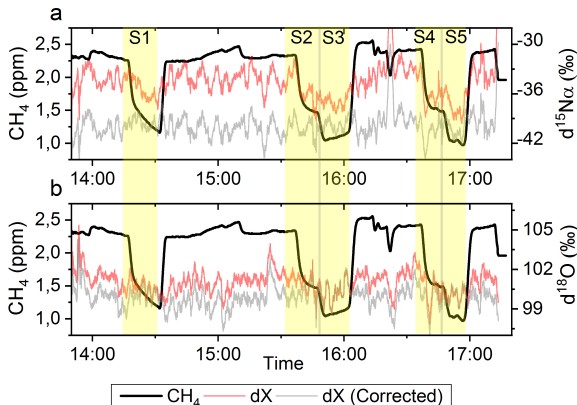

**Figure 5. a.** Measurements of $d^{15}N\alpha$ during exp. L in ‰. Red highlights a 100 s averaged measured values corrected for $O_2$, CO and $CO_2$ effects, while grey line indicates a 100 s average values corrected for all interference including $CH_4$. The black line shows the $CH_4$ level in ppm. **b.** Measurements of $d^{18}O$ from exp. L in ‰. Red highlights a 100 s averaged measured values corrected for $O_2$, CO and $CO_2$ effects, while grey line indicates a 100 s average values corrected for all interference including $CH_4$. The black line shows the $CH_4$ level in ppm.

**Effect of initial [$CH_4$]**

Exp. G, plotted in Figure 4d, spans [$CH_4$] in the range 1.4 - 3.8 ppm. Steps S1-S3 are highlighted to indicate the initial instability. The experiment showed high removal of methane at ambient concentrations.

The performance of the experimental setup has been investigated in the aforementioned experiments. The removal efficiencies can be increased by increasing $P_{IN}$ or [$Cl_2$] resulting in an increase in [Cl]. The negative correlation for [$CH_4$] is understand-
able as RE% is a relative value. As expected, the absolute amount of removed methane scales with the [$CH_4$].

### 3.1.1    $N_2O$ experimental results

In Figures 5a and 5b the effects on the isotopic signal of $d^{15}N\alpha$ and $d^{18}O$ of removing methane can be seen. The results are from experiment L, where a sofnocat trap had been installed to remove the CO formed by the $CH_4$ oxidation. By applying the trace-gas and matrix interference corrections described in Harris et al. (2020), it was found that the isotopic enrichments
remained stable through the oxidation (grey line). In comparison the uncorrected values (red) were several ‰ higher, however, the levels stabilized during the oxidation in accordance with the drop in methane, thus demonstrating the efficiency of the method. The stability of the corrected isotopic values across the experiment shows that the oxidation does not introduce other components that would interfere with the signal, which are not removed by the traps. Variations were observed in [$N_2O$] but are accounted for by vriations in [$Cl_2$].





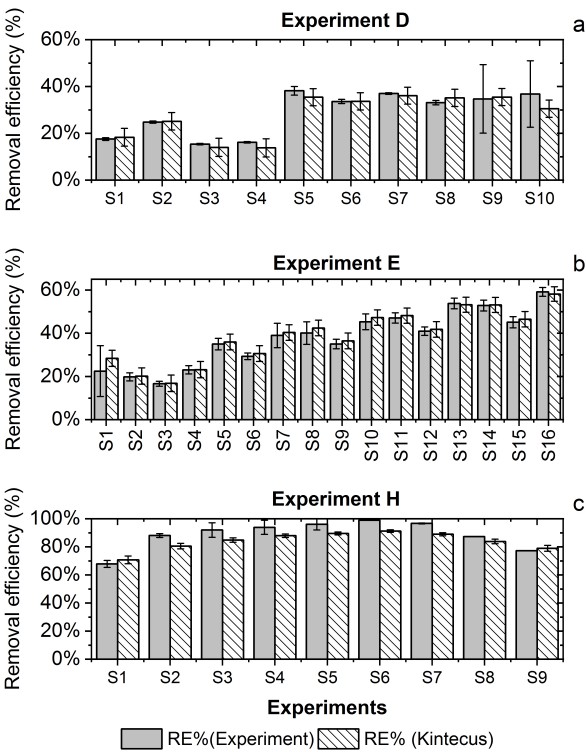

**Figure 6.** RE% for the steps of exp. D, F and H as found experimentally (White stripes) and by the model (Grey). **a.**Steps S2 and S6-S9 from exp. D were utilized to generate $J_{Cl_2}$ for the single tube system. **b.** Exp.E **c.** Exp.H

## 3.2 Model results

Parameters $a$ and $b$ in eq. (3) were determined from the experimental data. For the single tube system the values were fitted to steps S2 and S6-S9 from exp. D. Here two linear regimes were found, and were fitted by two sets of $a$ and $b$ constants. In this way we could describe the effect of the thermal management system used in later experiments.

The $J_{Cl_2}$ for the single tube systems is obtained from equations (C19) and (C20) (Figures C2c and C2a). These equations are used to calculate $J_{Cl_2}$ for exp. B, C and D. The comparison between the modeled and experimental efficiency is shown in Figure 6.

$J_{Cl_2}$ was determined using the same method. Exp. I is used to obtain model $J_{Cl_2}$ (Figures C2b-C2d and equation (C21).

In Figure 6c a comparison of experimental and model results are shown for exp. H, D and E. The model yields a good agreement with the experimental results. However, the model slightly underestimates RE% for most of the steps, which is also observed for the other experiments. The initial instability can also be seen for steps S1 and S2 depicted in Figure 6a. Problems due to overheating at high $P_{IN}$ are eliminated with the improved photochemical device resulting in a power effectiveness at





**Table 3.** Parameter ranges

| Parameter | Standard value | Range |
|---|---|---|
| $Cl_2$ | 50 ppm | 20-100 ppm |
| $CH_4$ | 2.04 ppm | 0.5-50 ppm |
| Residence time | 165 s | 40-400 s |
| $P_{IN}$ | 14.5 W | 9-31 W |
| $O_2$ | 10 % | |
| $N_2$ | 90 % | |

15 W of 0.6 % for the single tube to 9 % for the multiple tube system.

Overall, the simple model does a reasonable job of describing the experimental results although it underestimates the removal
efficiency. One issue is that the model does not do a good job of describing the effect of variations of initial methane concen-

210 trations in exp. G shown in Figure E1e.

Additional model runs are used to estimate $J_{Cl2}$ of experiments E and F, conducted with a modified device, cf. equations
(C22) and (C23)-(C24), respectively. It is clear that adjusting $J_{Cl_2}$ results in a model that more accurately fits the experimental
results.

### 3.2.1   Parameters simulated and compared with experimental results

Exp. I was chosen as the basis for the final simulation: three parameters are fixed and the fourth varies. The methane removal
efficiency, chlorine radical abundance, and the resulting abundance of [$CCl_4$] are determined. The standard values and the
ranges investigated can be seen in Table 3.  The resulting removal efficiencies as a function of each of the four parameters
Power input $P_{IN}$ (W), Residence time $t_R$ (s), [$Cl_2$] (ppm) and [$CH_4$] (ppm) are shown in Figure 7. The model results are
compared with the experimental results for the parameters $P_{IN}$ (W), $t_R$ (s) and chlorine mixing ratio (ppm), shown in Figures

4b, 4a and 4c, respectively. A good match in the observed response can be seen. The model is too insensitive to methane
concentration and fails to recreate the slope observed from the experimental results. The comparison between the model,
Figure 7d , and the experimental results, Figure 4d , shows that the model RE% scale is approximately one-tenth of that of the
experimental results. This may simply be due to the temperature depedence of the methane reaction rate. Simulations with an
increased $k_{Cl+CH_4}$ resulted in better agreement.

The corresponding $Cl_2$ photodissociation rates for the $P_{IN}$ in Figure 7a ranges from $4.04 \cdot 10^{-3}$ to $2.37 \cdot 10^{-2}$ photons s$^{-1}$
which is a good match with previous $J_{Cl_2}$ values found for a similar system. (Nilsson et al. (2009))

In addition to the RE%, [Cl] and [$CCl_4$] are also shown in the aforementioned Figures. Chlorinated side-products such as
$CH_3Cl$ and $CCl_4$ were investigated as another potential concern due to climate (Seinfeld and Pandis (2016)). Figure 7c shows
that an increase in $Cl_2$ concentrations increases the [$CCl_4$] production see Figures 7a , 7b and 7d. The amounts of carbon

tetrachloride formed are under a ppt for initial methane concentrations of tens of ppm i.e. yield on the order of less than $10^{-7}$.



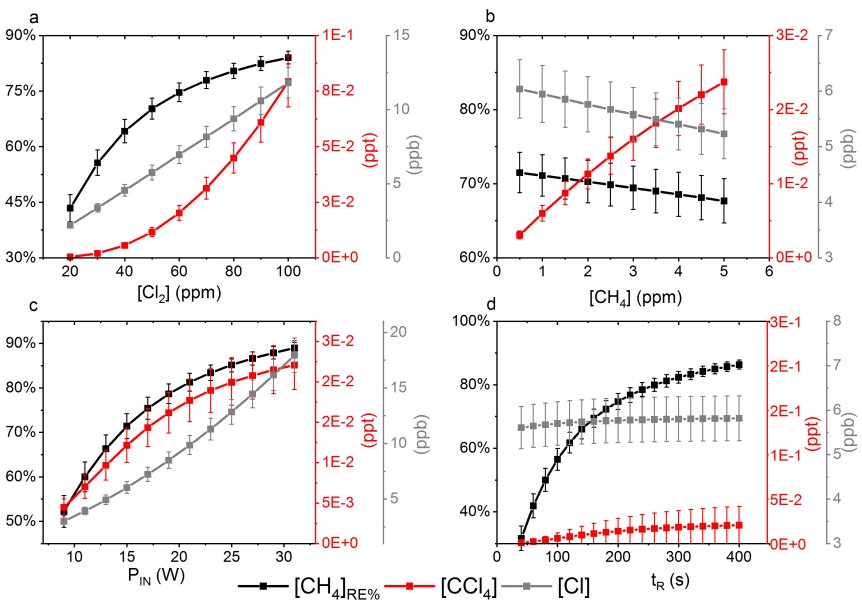

**Figure 7.** The Removal efficiency of methane depletion (Black), [CCl$_4$] (Red) and [Cl] (Grey) is shown in the Figures a-d. The four parameters are varied while the remaining parameters are kept at the standard parameter presented in Table 3. **a.** The [Cl$_2$] is varied. **b.** The initial [CH$_4$]is varied. **c.** The $J_{Cl_2}$ is varied. **d.** The $t_R$ is varied.

### 3.2.2 Side-reactions and products

The formation of HCl is unavoidable. As expected, the higher photolysis rate leads to more efficient methane-oxidation, and [HCl] crises accordingly. Therefore, scrubber technologies may be necessary. One method is to bubble the gas stream through a basic water solution e.g. NaOH. The NOx concentration in our experiments is insignificant these reactions have not been included in the model.

## 4 Conclusions

In this study we have described the design, improvement and performance of a process for continuously removing methane from an airstream. The system is based on the photolysis of chlorine gas using UV LED's to generate chlorine atoms. The performance of the setup was investigated on the basis of four variables; [CH$_4$], [Cl$_2$], photolysis rate, and $t_R$.

A model was built and used to describe the chemistry in more detail, and optimize the performance of the process. In addition, the model found that CCl$_4$ was produced at negligible levels.. The highest removal levels achieved experimentally at ambient methane levels were above 98 % which was maintained under stable conditions. A level above 99.5 % would be achievable by increasing the chlorine concentration or extending the photolysis time. The system was tested using N$_2$O isotope measure-



ments, a case where methane is known to interfere with mesurements of $d^{15}N\alpha$ and $d^{18}O$. With the inclusion of a sofnocat trap to control CO, the device was able to remove all interference from $H_2O$, $CO_2$, CO, and $CH_4$, thereby, thereby enabling accurate measurements of $[N_2O]$ and it's isotopically substituted analogues using the Picarro G5131-i.

We believe that researchers will be able to use this approach to continuously remove methane from a sample, thereby eliminating interference and improving accuracy.





## Appendix A: Proof-of-concept experiments - Preliminary experiments

Proof-of-concept experiments were conducted to investigate the feasibility of the proposed mechanism.

The ambient air standard was enriched in $Cl_2$ by in-situ production of $Cl_2$ through electrolysis of a saltwater mixture. Following that, the sample was photolyzed in a photochemical device generating Cl radicals. The resulting drop in methane was monitored with a Cavity Ring-Down Spectrometer, Picarro G1301.

The photochemical device comprised 28 LEDs (385 nm)(UV LED LAMP-VAOL-5EUV8T4) spaced evenly in a PVS plastic housing. The last set of experiments used a high-pressure Xenon lamp (ILC technology R100-IB) equipped with an optical filter at 335 nm. The resulting peak removal efficiencies for the preliminary experiments are presented in Table A1.

**Table A1.** Removal Efficiencies for the Preliminary Experiments

| Experimental setup (Date) | Highest Stable RE % | Initial [$CH_4$] (ppm) |
|---|---|---|
| A (17/4) | 68 % | 2 |
| A (23/4) | 67.75 % | 1.98 |
| A (24/4) | 76.48 % | 1.98 |
| B1 (26/4) | 78.52 % | 2 |
| B2 (30/4) | 80.16 % | 2 |
| C2 (26/5) | 98.20 %. | 2 |

The system yielded an average methane depletion of 86.63 % with a peak depletion at 98.2 %. Various parameters were changed throughout the experiments, and it was determined that the methane depletion is highly dependent on the flow, chlorine production, and light source. A better control of these parameters will yield higher and steadier removal of methane.

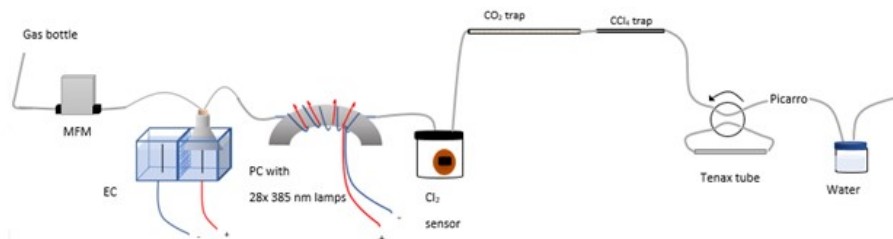

**Figure A1.** Experimental setup B2 with the inclusion of an activated carbon trap. Gas flask: Ambient air sample, MFM: mass flow meter, EC: electrolytic device, PC: photochemical device.

The experimental setup B2 is presented in Figure A1.



## The electrolytic device

The experimental setups presented in Table A1 uses an electrolytic device to produce chlorine gas. The electrolytic device is housed in a polycarbonate box. A Nafion membrane, Chemours, Nafion N234, is installed dividing the volume into two half
cells. Two electrodes are installed, and the two cells consist of two different solutions of NaCl in milli-Q water. The average concentration of NaCl is 1.3 M at the anodic site and 0.13 M at the cathodic site. The electrodes are carbon electrodes with a diameter of 2 mm and a length of 10 cm. On the anodic side $Cl_2$ is produced. (Pletcher and Walsh (2012)).

Anode reaction:

$$2Cl^- \rightarrow Cl_2(g) + 2e^- \tag{R11}$$

Cathode reaction:

$$2H^+ + 2e^- \rightarrow H_2(g) \tag{R12}$$

Overall reaction:

$$2NaCl + 2H_2O \rightarrow Cl_2 + H_2 + 2NaOH \tag{R13}$$

The presence of the membrane is essential due to its selectivity to cations. The membrane allows $Na^+$ ions move from the anode to the cathode and form NaOH. If the membrane was not present the NaOH would encounter $Cl_2$ and form hypochlorite.

$$2H_2O \rightarrow O_2(g) + 4H^+(aq) + 4e^- \tag{R14}$$

## The electrolysis chamber

In the experimental setups A to B2, Table A1, an electrolysis chamber is used to generate $Cl_2$, see Figure A1. The chamber
is made from PVC plastic. 30 holes were drilled, two holes for the 4 mm in diameter and 20 cm in length quartz tube and 28 holes for the LED (385 nm), UV LED LAMP-VAOL-5EUV8T4, diodes. The LED diodes are connected in parallel with a forward voltage and forward current. The max current is 20 mA for each LED, and the max voltage is 3.6 V. The same voltage runs through the LED and the current is multiplied by the number of lamps resulting in 0.480 A.

The chlorine gas is introduced into the gas stream by using a funnel above the anode. The water level is adjusted to yield
optimal condition for $Cl_2$ to get into the gas stream and avoid chlorine being deposited on the water surface or water getting sucked into the gas stream.


**Additional equipment**

The Picarro G1301 has a cavity pressure at 18.7 kPa, DAS temperature of 30.2 °C and cavity temperature of 45 °C.

The gas flask used has an ambient air combined with more stable concentrations of the investigated gases $CH_4$ (1.98 ppm),
$CO_2$ (376.1 ppm) and $H_2O$ (1.17473 % v).

The $Cl_2$ sensor used in all experiments is the PG610-CL2 model, Chlorine $Cl_2$ Gas Detector Gas Sound Light Vibration Alarm.

The sensor measures chlorine concentrations from 1-20 ppm. The sensor is placed in a 600 ml glass flask.

The general procedure is as follows:

- Prepare solutions

- Let the system stabilize

- Turn on the electrochemical device

- Let the $Cl_2$ concentration stabilize

- Turn on lamps

- Let the system stabilize to ensure a stable RE%.

305  - 10 min measurement with Tenax tube sampling (experiments B1 and B2)

- Turn off the light

- Let the system stabilize to the initial methane concentration

**Variations in the experimental setups**

Experimental setup A is the intial setup. Experimental setup B1 employed Tenax tube sampling for TD-GCMS measurements
of chlorinated species.

Experimental setup B2 follows the same procedure as B1, but with the addition of an activated carbon trap.

Experimental setup C1 uses a high-pressure Xenon lamp, ILC technology R100-IB. The Xenon lamp lights up the second
Photolyze Chamber (PC-2), which is equipped with an 8 mm in diameter and 20 cm in length quartz tube. The inner surface
of the cylinder is covered with aluminum foil to reflect the light coming in. The Xenon lamp emits light in wavelengths from
vacuum UV (200 nm) to infrared (Moore et al. (2009), therefore a 335 nm optical filter is installed.

At the Picarro G1301 outlet the two traps are used for trapping the gases hydrochloric acid, chlorine gas, and carbon dioxide.

Experimental setup C2 is similar to C1, however, the $Cl_2$ concentration is diluted to obtain the values above the fixed value
of 20 ppm. At the electrochemical device outlet a union tee divides the flow into two channels, one to the PC-2 and the other
to the sensor chamber. The flow at the outlet of the sensor chamber is measured by ADM Flow Meter to ensure a flow of
approximately 40-50 ml min$^{-1}$.





## Appendix B: Experimental setups ( $CH_4$ and $N_2O$)

**Table B1.** Table summarizing gas flask used in the experiments

| Flask name | $CH_4$ | $Cl_2$ | $N_2O$ | Matrix composition | Flow range |
|---|---|---|---|---|---|
| | (ppm) | (ppm) | (ppb) | | (ml min$^{-1}$) |
| A | 0 | $100 \pm 2.5$ | 0 | >99 % $N_2$ | 6-23 |
| B | $2.003 \pm 5 \cdot 10^{-4}$ | 0 | 0 | Atmospheric air | 1-29 |
| C | $78 \pm 2$ | 0 | 0 | 20.95 % $O_2$ + >79 % $N_2$ | 0.3-1.2 |
| D | 0 | 0 | 500 | Atmospheric air | 28-50 |

The photochamber for High Pressure Xenon Lamp (HPXL)-setup uses a quartz tube with dimensions (20 cm in length, 1/2 inches(12.7 mm) in outer diameter) placed in a cylinder coated with aluminium.

The Photochemical Device (PD), Figure B2, for later experiments, Figure B1b - B1f, consists of 420 LED diodes at 365 nm peak wavelength. The LED diodes run in a parallel circuit with a forward voltage and forward current (from positive to negative). The max current is 13.2 mA for each LED, and the max voltage is 3.8 V. The same voltage runs through the LEDs, resulting in a total current across the system of 5.5 A.

The difference between the two similar setups STH-PD and STH-PD-MFC are illustrated in Figures B1b and B1c, respectively. Here the forward pressure valve is exchanged with a mass flow controller to allow for a smaller and more stable level of vent flow. The quartz tube of the previous experiments is substituted with seven smaller quartz tubes for MTH-PD setup to yield a longer $t_R$.

The seven quartz tubes are structured in a hexagonal shape for optimal packing and consist of five 8.33 mm in outer diameter with a 20 cm length tubes, and two tubes with an outer diameter of 8.00 mm and a length of 25 cm. The extra 5 cm of the two last tubes were outside the photolysis chamber for the purpose of connecting with the setup. The tubes were connected in series via Tygon tubes, Tygon R3603, of length 5 cm. The insides of these tubes were coated with Krytox, GPL-206 Grease Lubricant O-Rings SCUBA, in order to stop the consumption of $Cl_2$.

### B1 Experimental procedure

– Tune the desired flow from flask C for methane and mix it with a flow from flask B equal to the desired flow plus the intended flow from flask A.

– Let the system stabilize.

– Add the desired flow of chlorine from flask A, by adjusting the pressure at the flask.

– Reduce the flow from flask B by an equal amount to get the desired mixing ratio.





**Figure B1. a.** High Pressure Xenon Lamp (HPXL)-setup. This setup was utilized for exp. A. XL: High pressure Xenon lamp. See Table B1 for gas flask supply. ACT: Activated Carbon Trap. MFM: Mass Flow Meter. MFC: Mass Flow Controller. **b.** Single Tube hexagonal Photochemical Device (STH-PD-PC) - setup. Setup was utilized during exp. B. See Table B1 for gas flask supply. ACT: Activated Carbon Trap. MFM: Mass Flow Meter. MFC: Mass Flow Controller. **c.** Single Tube Hexagonal Photochemical device with MFC. (STH-PD-MFC) setup. Setup used in exp. C, D and E. See Table B1 for gas flask supply. ACT: Activated Carbon Trap. MFM: Mass Flow Meter. MFC: Mass Flow Controller. **d.** Multiple Tube Hexagonal Photochmical Device (MTH-PD) -setup. Setup used in exp. F-I. See Table B1 for gas flask supply. ACT: Activated Carbon Trap. MFM: Mass Flow Meter. MFC: Mass Flow Controller. **e.** Multiple Tube Hexagonal Photochmical Device (MTH-PD) with co-measurements of $N_2O$. The setup was used for exp. J, K and L in combination with the second $N_2O$ setup. See 2 for gas flask supply. ACT: Activated Carbon Trap. MFM: Mass Flow Meter. MFC: MassFlow Controller **f.** Multiple Tube Hexagonal Photochmical Device (MTH-PD) with co-measurements of $N_2O$ and a sofnocat trap installed. This setup introduces a sofnocat trap to the setup immediately following the ascarite trap. The setup was used in exp. L in combination with the first $N_2O$ setup. See 2 for gas flask supply. ACT: Activated Carbon Trap. MFM: Mass Flow Meter. MFC: MassFlow Controller





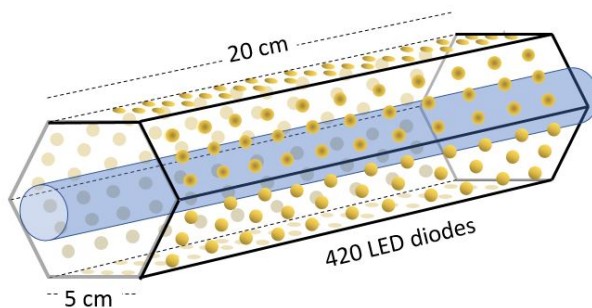

**Figure B2.** Hexagonal photochemical device consisting of connected circuit boards of 420 LED at 365 nm

- Let the system stabilize and confirm that the resulting total flow fits with the expected. Make sure the chlorine value can be read on the chlorine sensor.

– When a stable methane level has been run for sufficient time, turn on the photochemical device.

- Let the system stabilize to ensure a stable methane RE%.

- Turn off the light.

- Let the system stabilize to the initial methane concentration before the light was turned on.

## B2  N$_2$O experiments

Experiments were conducted with the Picarro model G5131-i, which is used to measure N$_2$O mixing ratio and isotopic abundance. The purpose of the experiments was to confirm that the illumination did not affect N$_2$O. The experimental setups are shown in Figures B1e and B1f. The difference between the two was the inclusion of a sofnocat trap for oxidizing the formed CO (Harris et al. (2020)). The sofnocat trap was prepared with 1.25 g of sofnocat contained in a 6.4 mm diameter tube of length 8 cm and kept in place by glass wool. The trap was installed to prevent effects on the N$_2$O isotope signal from CO. The

installation of this trap after the CO$_2$ trap allowed us to measure the amount of CO present. The technical air from flask C was exchanged with a technical air mix with a 509 ppb [N$_2$O], allowing for dilution to ambient level. The flow ratio between the three different gases was regulated to maintain a mixing ratio of 330 ppb N$_2$O, 2.4 ppm CH$_4$ and 33 ppm Cl$_2$. Power supply to the lamp was constant at 4.8 V and 5.0 A, and $t_R$ in the chamber was varied between 86, 117 and 145 s.





## Appendix C: Theoretical models

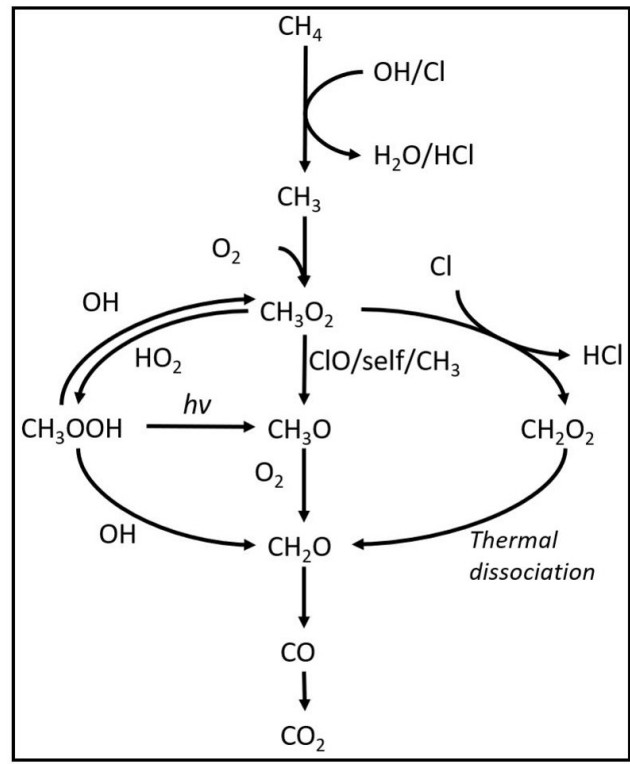

**Figure C1.** Reaction scheme for the oxidation of methane to $CO_2$. $CH_3O_2$ self reactions lead to the formation of $CH_3O$

The model is made with the program Kintecus, version 6.8, (Ianni (2012)). The model was developed by describing the relevant reactions with rates for chlorine radical production/removal and formation of chlorinated species. The model was kept as simple as possible while still including key reactions. The reactions and their rates used in the model are found in Tables E1 - E3. A simplified reaction-scheme is shown in Figure C1. The experiments are modeled by choosing the concentrations of both the initial and external concentration of used species and the $t_R$ within the chamber. A continuous flow was modeled

by setting the initial and external concentrations of gases flowing through the chamber to the same value. This is done for the gases $Cl_2$, $CH_4$, $N_2$ and $O_2$.

The physical parameters are fixed as well. The temperature at 298 K, starting integration time to $10^{-6}$ s (starting step for the integrated model), maximum integration time to 1 s, simulation length equal to $t_R$ plus 5 s, the accuracy of digits to $10^{-4}$. Furthermore, the energy unit kcal was selected, and the unit of concentration selected to be $\mathrm{molecule\ cm^{-3}}$.





## C1 Radical wall reactions

As described in the main article a set of radical terminating reactions was incorporated into the mode. The wall reaction rates were estimated based on the diffusion rate of the radicals and the diffusion length. The diffusion length is calculated as the average distance from the wall. Because two different sizes of tubes were used throughout the experiments the wall reactions reflects that. The diffusion length and the diffusion rate is given in eq. (C1) and eq. (C2), respectively.

$$l = 2 \cdot (D \cdot t)^{0.5} \tag{C1}$$

Where D is the diffusion constant and $t$ is time.

$$k = \frac{1}{t} = \frac{4 \cdot D}{l^2} \tag{C2}$$

The distance, $l$, is defined as the average distance from the wall, which can alternatively be written as $l = r\text{-}dc$. Where $r$ is the radius of the tube, and $dc$ is distance from a random particle in the cylinder to the center of the circle of the cylinder. Finding the average distance to the wall of an infinite number of randomly located particles in the cylinder can be accomplished by solving the equation (C3). The result of eq. (C4) is used to calculate the resulting diffusion rate with the inclusion of the average distance from the walls of the tube, which is defined in eq. (C5).

$$\frac{1}{2}A = \int_0^{dc} 2r \cdot \pi dr \tag{C3}$$

Where r is the radius and $A = r^2 \cdot \pi$ is the area.

$$dc = \left(\frac{A}{2\pi}\right)^{\frac{1}{2}} = \frac{r}{\sqrt{2}} \tag{C4}$$

$$k = \frac{4 \cdot D}{r - \frac{r}{\sqrt{2}}} \tag{C5}$$

The diffusion constants, diffusion lengths, and estimated wall reaction rates are shown in Table E3.

## C2 $J_{Cl_2}$ estimation

### C2.1 First approach

The first approach is to fit $J_{Cl_2}$ in the model to regenerate the observed removal efficiencies from experimental results. These fits were only produced for experiments investigating the effect of $P_{in}$. The resulting $J_{Cl_2}$ was related to $P_{in}$ via the effective power to light conversion based on the absorption cross-section of $Cl_2$ and the wavelength distribution of the LEDs. $J_{Cl_2}$ was determined in this manner, once for the single tube systems and once for the multiple tube systems. The photolysis rate $J$ in the units of $\text{photons}^{-1}$ can be determined by equation C6.

$$J_{Cl_2} = \int \sigma(\lambda, T) \cdot \phi(\lambda, T) \cdot I(\lambda, W) d\lambda \tag{C6}$$





**Table C1.** Radical Wall reaction parameters. *Diffusion coefficient is estimated from $D_{HO-Air}$ and $D_{HO_2-He}$, ** The Chapman and Cowling diffusion model was used to estimate the diffusion constant.

| Setup | Reaction | Diffusion constant (cm$^2$s$^{-1}$) | Reference | Diffusion length (cm) | Wall-reaction rate (s$^{-1}$) |
|---|---|---|---|---|---|
| Single tube | $Cl \rightarrow \frac{1}{2}Cl_2$ | 0.260 | Judeikis and Wun (1978) | 0.146 | 1.2E+02 |
| Multiple tube | | | | 0.091 | 4.8E+01 |
| Single tube | $ClO \rightarrow \frac{1}{2}Cl_2 + \frac{1}{2}O_2$ | 0.184 | Seinfeld and Pandis (2016)** | 0.146 | 8.8E+01 |
| Multiple tube | | | | 0.091 | 3.4E+01 |
| Single tube | $OH \rightarrow \frac{1}{2}H_2O + \frac{1}{4}O_2$ | 0.217 | Ivanov et al. (2007) | 0.146 | 1.0E+02 |
| Multiple tube | | | | 0.091 | 4.0E+01 |
| Single tube | $HO_2 \rightarrow \frac{1}{2}H_2O + \frac{3}{4}O_2$ | 0.139 | Ivanov et al. (2007)* | 0.146 | 6.7E+01 |
| Multiple tube | | | | 0.091 | 2.6E+01 |

Where $\sigma(\lambda, T)$ is the wavelength dependent cross section of $Cl_2$ with the unit $cm^2$ molecule$^{-1}$, $\phi(\lambda, T)$ is the quantum yield, and $I(\lambda, W)$ is the spectral actinic flux density in photons $cm^{-2}$ s$^{-1}$ nm$^{-1}$. The cross-section of chlorine dissociation in the range 250-550 nm is defined by C7. (Burkholder et al. (2020))

$$\sigma(\lambda, T) = 10^{-20}(tanh(\frac{402.7}{T}))^{0.5}.$$
$$(27.3 \cdot e^{-99.0 \cdot (tanh(\frac{402.7}{T})) \cdot (ln(\frac{329.5}{\lambda}))^2}$$
$$+ 0.932 \cdot e^{-91.5 \cdot tanh(\frac{402.7}{T}) \cdot (ln(\frac{406.5}{\lambda}))^2}) \tag{C7}$$

Where T is the temperature, and $\lambda$ is the wavelength in nm.

$$I(\lambda, W) = \frac{P(\lambda, W) \cdot D(\lambda) \cdot l}{V} \tag{C8}$$

The actinic flux, eq. (C8)) is a function dependent on the power output ($P(\lambda, W)$ from eq. (C9), the distribution (D($\lambda$)) from eq. (C11) and the tube volume (V).

$$P(\lambda, W) = Eff(W) \cdot \frac{\lambda}{hc} \tag{C9}$$

Where $h$ is planks constant, and $c$ is the speed of light.

It was observed that the photolysis rate did not scale linearly with $P_{IN}$ which may be due to variation of the efficiency of the lamp with applied current and operating temperature. This effect was found to be sufficiently described by a linear fit and is defined as Eff(W).

$$Eff(W) = a \cdot W + b \tag{C10}$$

Where W is the power supplied to the diodes.





The function C10 accounts for additional variations such as effects due to temperature, the cross-section area of the quartz tube, the conductance of the photochamber and the quality of the distribution fit. The constants $a$ and $b$ were found to vary according to changes in these parameters.

The photon output, eq. (C9), from the LED diodes was assumed to follow a normal distribution. For this distribution shown in equation (C11), we assumed a center value of 365 nm and FWHM of 10 nm. The distribution, eq. (C11), has units of $nm^{-1}$.

$$D(\lambda) = \frac{1}{(10\text{nm} \cdot (2\pi)^{0.5} \cdot e^{-0.5 \cdot (\frac{\lambda - 365\text{nm}}{10\text{nm}})^2}} \tag{C11}$$

The photolysis rate could then be calculated by eq. (C6) across 250-500 nm at 298 K. Values for the constants $a$ and $b$ from eq. (C10) are fitted in the model to match the experiment.

### C2.2   Second approach

A second approach for estimating $J_{Cl_2}$ and relating it to $P_{IN}$ was used. This method estimated $J_{Cl_2}$ using simplified kinetics and relating it to power via the same method as the model derived $J_{Cl_2}$. Exp. F reflects the single tube system while exp. I reflects the optimized multiple tubes setup. Four main reactions, R15 - R18, are considered in the simple kinetic model

$$\text{Cl}_2 + h\nu \xrightarrow{J_{Cl_2}} 2\text{Cl} \tag{R15}$$

$$\text{Cl} + \text{CH}_4 \xrightarrow{k_{Cl+CH_4}} \text{CH}_3 + \text{HCl} \tag{R16}$$

$$[k_{Cl+CH_4} = 1.07 \cdot 10^{-13} \cdot \text{molecules}^{-1}\text{cm}^3\text{s}^{-1}]$$

$$\text{Cl} + \text{Cl} + M \xrightarrow{k_{self}} \text{Cl}_2 + M \tag{R17}$$

$$[kself = 1.24 \cdot 10^{-32} \cdot \text{molecules}^{-2}\text{cm}^6\text{s}^{-1} \cdot [M]]$$

$$\text{Cl} \xrightarrow{k_{wall}} \frac{1}{2}\text{Cl}_2 \tag{R18}$$

$$[k_{wall} = 124.5\text{s}^{-1} \text{ or } 48.9\text{s}^{-1}]$$

The Cl radicals are consumed at a fast rate, therefore, steady state approximation for Cl has been assumed.

$$\begin{aligned}
\frac{d[Cl]}{dt} = {} & 2 \cdot J_{Cl_2}[Cl_2] - (2 \cdot k_{self} \cdot [Cl]^2 \\
& + k_{Cl+CH_4} \cdot [CH_4] \cdot [Cl] + k_{wall} \cdot [Cl]) = 0
\end{aligned} \tag{C12}$$

The photolyze rate for the kinetic calculation is, thereby, defined in equation C13

$$J_{kin} = \frac{2 \cdot k_{self} \cdot [Cl]^2 + k_{Cl+CH_4}[CH_4][Cl] + k_{wall}[Cl]}{2 \cdot [Cl_2]} \tag{C13}$$

The photolysis rate is calculated from an estimated [Cl] concentration. This was achieved by assuming that the methane concentration would follow an exponential decay with time, equation (C14). The estimated [Cl] is expressed in equation (C15).

$$[\text{CH}_4]_t = [\text{CH}_4]_0 \cdot exp(-k_{\text{Cl+CH}_4} \cdot [Cl] \cdot t) \tag{C14}$$





Where $[CH_4]_t$ is the methane concentration at time $t$, while $[CH_4]_0$ is the initial concentration.

$$[Cl] = ln(\frac{1}{1-RE})/(k_{Cl+CH_4} \cdot t) \tag{C15}$$

The values for $J_{kin}$ are generated by inserting the experimental values of $[Cl_2]$, $[CH_4]$ and the estimated value of $[Cl]$ into eq.( C13).

The distribution function $D(\lambda)$ from equation (C11) can be used in combination with the cross-section to determine the scale factor $J_{scale}$.

$$J_{scale}(\lambda,T) = \int\limits_{250nm}^{500nm} \frac{\lambda}{hc} \cdot \sigma(\lambda,T) \cdot \frac{l \cdot D(\lambda)}{V} d\lambda \tag{C16}$$

The value of $J_{scale}$ $j^{-1}$ is calculated from the overlap integral between $\sigma(\lambda,T)$ and the emitted photon distribution.

$l$ is the pathlength across the tube/tubes in cm, and $V$ is the volume of the tube/tubes in ml. $\lambda$ is the wavelength in nm, $h$
is Planck's constant and $c$ is the speed of light. Values for the constants $a$ and $b$ from eq. (C17) are then fitted to match the photolysis rate in eq. (C18) with the photolysis rate found from the Kintecus model.

$$P_{eff}(P_{IN}) = (a \cdot P_{IN} + b) \cdot P_{IN}$$
$$= Eff(kin) \cdot P_{IN} = \frac{J_{Kin}}{J_{Scale} \cdot P_{IN}} \tag{C17}$$

Where $P_{eff}$ is the effective power, and the constants $a$ and $b$ are setup-dependent constants.

From the effective power output the photolysis rate $J_{Cl_2}$ could be calculated by multiplying $P_{eff}$ with $J_{scale}$.

$$J_{Cl_2}(P_{IN}) = P_{eff}(P_{IN}) \cdot J_{scale} \tag{C18}$$

## C3  $J_{Cl_2}$ fitted to collected data

The $J_{Cl_2}$ is fit to the data collected for some of the experimental steps for exp. F and I to determine the values for the constants $a$ and $b$.

Exp. F is the single tube system and exp. I is the optimized multiple tube system. From the fitted $a$, $b$ and calculated $J_{scale}$ the
photolysis rate could be calculated for the other experiments.

**Single tube systems**

$J_{Cl_2}$ values are generated on the basis of exp. D2 and D6-D9. The efficiency of $p_{IN}$ is generated from the $J_{Cl_2}$ model. A correlation between effectiveness (%) and experimental $p_{IN}$ (W) is shown in Figure C2a as well as the correlation with the $J_{Cl_2}$ (Kintecus) values in Figure C2c.

The $J_{Cl_2}$ dependence on the $p_{IN}$ (W) for the single tube system, exp. B, C and D, is given by the equations C19 and C20. The equations incorporate a decrease in efficiency of $p_{IN}$ at higher levels due to overheating of the chamber as seen in Figure D2c.

$$J = 2.59 \cdot 10^{-2} \cdot (2.3 \cdot 10^{-4} \cdot (P_{IN})^2 + 2.99 \cdot 10^{-3} \cdot P_{IN})$$
$$if(P_{IN} > 14.67W \tag{C19}$$





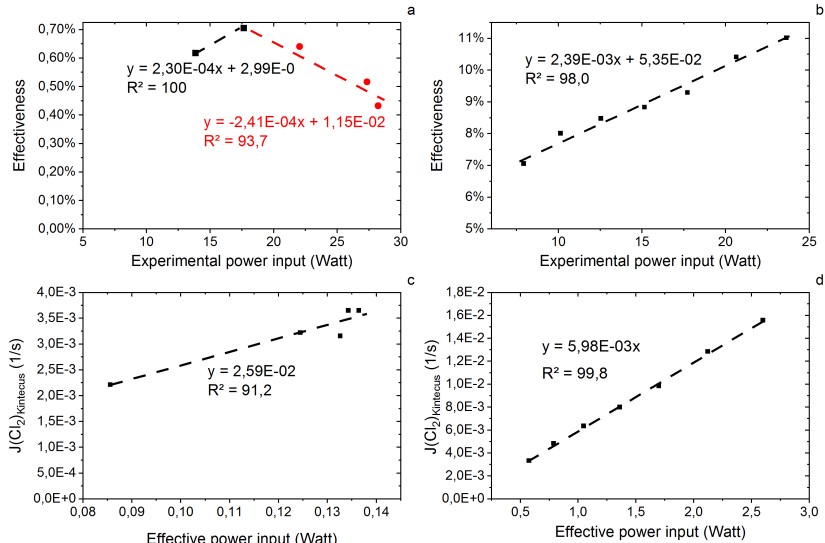

**Figure C2. a.** Effectiveness as a function of experimental power ínput for exp. D. The correlation is used for calculating the effective $P_{IN}$ for single tube experiments. **b.** Effectiveness as a function of experimental $P_{IN}$ for exp. I. The correlation is used for calculating the effective power for multiple tubes experiments. **c.** Kinecus obtained $J_{Cl_2}$ as a function of the effective $p_{IN}$ for exp. D. The effective $p_{IN}$ is calculated from Figure C2a. The combination of the Figure with Figure C2a is used to calculate the $J_{Cl_2}$ for single tube experiments by Eq. (C19) and (C20). **d.** Kinecus obtained $J_{Cl_2}$ as a function of the effective $p_{IN}$ exp. I. The effective $p_{IN}$ is calculated from Figure C2b. The combination of the Figure with Figure C2b is used to calculate the $J_{Cl_2}$ for Multiple tubes experiments by Eq. (C21)

$$J = \left(2.59E \cdot 10^{-2} \cdot \left(-2.41 \cdot 10^{-4} \cdot (P_{IN})^2 + 1.15 \cdot 10^{-2} \cdot P_{IN}\right)\right.$$
$$if (P_{IN} < 14.67W \tag{C20}$$

The comparison between modeled and experimental efficiency for single tube experiment is seen in Figures E1a and E1b.

**Multiple tubes systems**

$J_{Cl_2}$ is generated in the same manner as the experiment of results with multiple tubes. Here exp. I is used to obtain model $J_{Cl_2}$-values. (Figures C2b and C2d).

$$J = 5.98 \cdot 10^{-3} \cdot \left(2.39 \cdot 10^{-3} \cdot (P_{IN})^2 + 5.35E \cdot 10^{-2} \cdot P_{IN}\right) \tag{C21}$$

The overheating at high $p_{IN}$ is eliminated with the improved photochemical device. This is also apparent when comparing the effectiveness, which is approximately 9 % for the multiple tube configuration, Figure C2b, and approximately 0.6 % for the single tube system, Figure C2a, at the same $p_{IN}$ at 15 W. Figure E1e and E1f shows the comparison for exp. G and H,





respectively.

**Exp. E and F**

Some experiments can't be related to the relations presented for the single and multiple tube systems. This is due to the optimization done on the photochemical device. A second approach with additional kinetic calculations is, therefore, used to estimate the $J_{Cl_2}$ of these two experiments. The effectiveness of exp. E is shown in equation (C22).

$$P_{eff}(P_{IN}) = P_{IN}$$
$$(-4.35 \cdot 10^{-3} \cdot P_{IN}$$
$$+ 3.26 \cdot 10^{-2}) \tag{C22}$$

In the same manner the effectiveness of exp. F in shown in equations C23 and C24.

$$P_{eff}(P_{IN}) = P_{IN} \cdot (6.80 \cdot 10^{-4} \cdot$$
$$P_{IN} + 4.36 \cdot 10^{-2})$$
$$if\, P_{IN} > 14.31 W \tag{C23}$$

$$P_{eff}(P_{IN}) = P_{IN} \cdot (-1.57 \cdot 10^{-3} \cdot$$
$$P_{IN} + 7.58 \cdot 10^{-2})$$
$$if\, P_{IN} < 14.31 W \tag{C24}$$





## Appendix D: Settings and experimental results

**Table D1.** Data for exp. A-D. Columns: Experimental steps, [CH$_4$] (ppm), [Cl$_2$] (ppm), Residence time $t_R$ (s), Power input $p_{IN}$ (W) and the Resulting removal efficiency in %.

\* : The $p_{IN}$ of the xenon lamp was not varied nor determined.

| Experiment (#) | CH$_4$ (ppm) | Cl$_2$ (ppm) | Residence time (s) | Power (W) | Removal efficiency (%) |
|---|---|---|---|---|---|
| Exp.A | | | | | |
| 1 | 3.2729 ± 7E-04 | 16.7 ± 1. | 62.4 ± 1.6 | * | 0.01 ± 0.04 |
| 2 | 2.8327 ± 1.5E-03 | 25 ± 2 | 62 .2 ± 1.6 | * | 6.40 ± 0.4 |
| 3 | 2.3769 ± 9E-04 | 50 ± 5 | 61.8 ± 1.6 | * | 27.29 ± 0.4 |
| 4 | 2.9367 ± 3E-03 | 92 ± 11 | 62.1 ± 1.6 | * | 2.69 ± 0.4 |
| Exp.B | | | | | |
| 1 | 3.6391 ± 5E-03 | 16.7 ± 1.5 | 60.7 ± 1.6 | 17.43 ± 0.03 | 2.01 ± 0.5 |
| 2 | 3.6598 ± 1E-03 | 16.7 ± 1.5 | 60.9 ± 1.5 | 26.13 ± 0.04 | 6.52 ± 0.2 |
| 3 | 3.7069 ± 1.2E-03 | 16.7 ± 1.5 | 62.2 ± 1.6 | 9.91 ± 0.03 | 6.87 ± 0.2 |
| 4 | 3.7268 ± 6E-03 | 16.7 ± 1.5 | 62.4 ± 1.6 | 22.09 ± 0.04 | 10.96 ± 0.6 |
| 5 | 3.919 ± 1.5E-02 | 50 ± 5 | 61.0 ± 1.5 | 9.92 ± 0.03 | 23.91 ± 0.8 |
| 6 | 3.945 ± 1.4E-02 | 50 ± 5 | 61.4 ± 1.5 | 16.59 ± 0.03 | 32.69 ± 0.6 |
| Exp.C | | | | | |
| 1 | 3.955 ± 2E-02 | 50 ± 5 | 129 ± 4 | 9.63 ± 0.03 | 44.51 ± 5.2 |
| 2 | 3.957 ± 6E-02 | 50 ± 5 | 41.4 ± 1.0 | 9.63 ± 0.03 | 18.90 ± 1.3 |
| 3 | 4.0301 ± 5E-03 | 50 ± 5 | 41.4 ± 1.0 | 17.30 ± 0.03 | 27.83 ± 1.0 |
| 4 | 3.986 ± 1.0E-02 | 50 ± 5 | 128 ± 4.7 | 17.30 ± 0.03 | 60.6 ± 8.7 |
| Exp.D | | | | | |
| 1 | 3.5395 ± 5E-03 | 32 ± 3 | 62.5 ± 1.5 | 13.38 ± 0.03 | 17.55 ± 0.6 |
| 2 | 3.531 ± 1.5E-02 | 32 ± 3 | 85 ± 14 | 13.85 ± 0.03 | 24.82 ± 0.5 |
| 3 | 3.5570 ± 8E-03 | 32 ± 3 | 43.4 ± 1.1 | 13.73 ± 0.03 | 15.39 ± 0.3 |
| 4 | 3.5405 ± 9E-03 | 32 ± 3 | 43.0 ± 1.0 | 13.65 ± 0.03 | 6.18 ± 0.3 |
| 5 | 3.526 ± 4E-02 | 32 ± 3 | 135 ± 58 | 13.63 ± 0.03 | 38.19 ± 1.8 |
| 6 | 3.5261 ± 3E-03 | 32 ±3 | 84 ± 2.6 | 17.65 ± 0.03 | 33.61 ± 0.9 |
| 7 | 3.564 ± 5E-02 | 32 ± 3 | 83 ± 2.2 | 22.05 ± 0.04 | 37.02 ± 1.3 |
| 8 | 3.567 ± 5E-02 | 32 ± 3 | 83 ± 2.2 | 28.22 ± 0.04 | 33.09 ± 1.5 |
| 9 | 3.5729 ± 3E-03 | 32 ± 3 | 83 ± 2.2 | 27.34 ± 0.04 | 34.72 ± 15 |
| 10 | 3.5447 ± 7E-03 | 32 ± 3 | 79 ± 2.1 | 33.10 ± 0.04 | 36.83 ± 14 |





**Table D2.** Data for exp. E-F. Columns: Experimental steps, [CH$_4$] (ppm), [Cl$_2$] (ppm), Residence time $t_R$ (s), Power input $p_{IN}$ (W) and the Resulting removal efficiency in %.

| Experiment (#) | CH$_4$ (ppm) | Cl$_2$ (ppm) | Residence time (s) | Power (W) | Removal efficiency (%) |
|---|---|---|---|---|---|
| Exp.E | | | | | |
| 1 | 3.3805 ± 6E-03 | 30 ± 3 | 62.7 ± 1.6 | 13.39 ± 0.03 | 22.46 ± 12 |
| 2 | 3.3984 ± 2E-03 | 20 ± 2 | 61.6 ± 1.7 | 13.36 ± 0.03 | 19.80 ± 1.9 |
| 3 | 3.3947 ± 3E-03 | 20 ± 2 | 61.4 ± 1.5 | 9.89 ± 0.03 | 16.65 ± 1.2 |
| 4 | 3.4014 ± 9E-04 | 20 ± 2 | 61.4 ± 1.5 | 17.50 ± 0.03 | 23.09 ± 1.9 |
| 5 | 3.3282 ± 5E-03 | 40 ± 4 | 61.0 ± 1.5 | 13.43 ± 0.03 | 34.99 ± 2.7 |
| 6 | 3.3309 ± 5E-03 | 40 ± 4 | 61.1 ± 1.6 | 9.92 ± 0.03 | 29.33 ± 1.6 |
| 7 | 3.3312 ± 4E-03 | 40 ± 4 | 61.0 ± 1.6 | 17.47 ± 0.03 | 38.99 ± 5.6 |
| 8 | 3.4096 ± 6E-03 | 50 ± 5 | 60.7 ± 1.5 | 13.43 ± 0.03 | 40.12 ± 5.2 |
| 9 | 3.4444 ± 3E-03 | 50 ± 5 | 60.6 ± 1.5 | 9.90 ± 0.03 | 35.05 ± 2.2 |
| 10 | 3.4377 ± 3E-03 | 50 ± 5 | 60.6 ± 1.5 | 17.49 ± 0.03 | 45.32 ± 3.7 |
| 11 | 3.3575 ± 5E-03 | 60 ± 6 | 60.4 ± 1.5 | 13.43 ± 0.03 | 47.06 ± 2.4 |
| 12 | 3.3800 ± 7E-03 | 60 ± 6 | 60.4 ± 1.5 | 9.90 ± 0.03 | 41.00 ± 1.9 |
| 13 | 3.3604 ± 3E-03 | 60 ± 6 | 60.3 ± 1.6 | 17.49 ± 0.03 | 53.80 ± 2.4 |
| 14 | 3.4122 ± 3E-03 | 70 ± 7 | 60.1 ± 1.6 | 13.43 ± 0.03 | 52.86 ± 2.5 |
| 15 | 3.4414 ± 1.5E-03 | 70 ± 7 | 60.0 ± 1.6 | 9.90 ± 0.03 | 45.12 ± 2.6 |
| 16 | 3.4566 ± 8E-03 | 70 ± 7 | 59.8 ± 1.6 | 17.49 ± 0.03 | 59.13 ± 2.0 |
| Exp.F | | | | | |
| 1 | 3.5176 ± 6E-03 | 50 ± 5 | 162 ± 3.4 | 6.75 ± 0.02 | 27.99 ± 0.3 |
| 2 | 3.5475 ± 1.1E-03 | 50 ± 5 | 162 ± 3.4 | 9.74 ± 0.02 | 37.34 ± 0.3 |
| 3 | 3.5668 ± 1.8E-03 | 50 ± 5 | 161 ± 3.4 | 12.17 ± 0.03 | 46.83 ± 0.1 |
| 4 | 3.5920 ± 1.0E-03 | 50 ± 5 | 161 ± 3.4 | 14.63 ± 0.03 | 53.77 ± 0.1 |
| 5 | 3.6162 ± 1.9E-03 | 50 ± 5 | 161 ± 3.3 | 17.18 ± 0.03 | 55.24 ± 0.2 |
| 6 | 3.6425 ± 3E-03 | 50 ± 5 | 160 ± 3.4 | 19.73 ± 0.03 | 57.96 ± 0.3 |
| 7 | 3.6592 ± 1.2E-03 | 50 ± 5 | 160 ± 3.4 | 22.32 ± 0.03 | 59.22 ± 0.3 |

**D1   CH$_4$ experimental results**

In Tables D1 - D3 the four varying parameters; [CH$_4$]$_{Initial}$, [Cl$_2$], $t_R$ and $P_{IN}$ are presented for each experiment alongside the resulting RE% . Table D4 summarizes the experiments done in the study.



**Table D3.** Data from exp. G - I. Columns: Experimental steps, [CH$_4$] in ppm, [Cl$_2$] in ppm, Residence time in seconds, Power in watts and the Resulting removal efficiency in %. **: The [CH$_4$] values are calculated based on trend fitting

| Experiment (#) | CH$_4$ (ppm) | Cl$_2$ (ppm) | Residence time (s) | Power (W) | Removal efficiency (%) |
|---|---|---|---|---|---|
| **Exp.G** | | | | | |
| 1 | 3.5594 ± 1.7E-03 | 50 ± 5 | 167 ± 3.5 | 14.46 ± 0.03 | 47.40 ± 1.2 |
| 2 | 3.2339 ± 1.3E-03 | 50 ± 5 | 168 ± 3.5 | 14.49 ± 0.03 | 54.73 ± 0.5 |
| 3 | 2.9339 ± 9E-04 | 50 ± 5 | 168 ± 3.5 | 14.56 ± 0.03 | 60.77 ± 0.5 |
| 4 | 2.684 ± 4E-02 | 50 ± 5 | 167 ± 3.6 | 14.60 ± 0.03 | 66.22 ± 0.6 |
| 5 | 2.2942 ± 3E-03 | 50 ± 5 | 164 ± 3.4 | 14.63 ± 0.03 | 69.64 ± 0.3 |
| 6 | 1.9817 ± 6E-04 | 50 ± 5 | 164 ± 3.5 | 14.46 ± 0.03 | 72.01 ± 0.4 |
| 7 | 1.6982 ± 7E-04 | 50 ± 5 | 166 ± 3.4 | 14.46 ± 0.03 | 74.06 ± 0.6 |
| 8 | 1.3899 ± 3E-04 | 50 ± 5 | 163 ± 3.5 | 14.46 ± 0.03 | 77.19 ± 0.7 |
| 9 | 3.8333 ± 7E-03 | 50 ± 5 | 162 ± 3.4 | 14.70 ± 0.03 | 60.32 ± 0.4 |
| 10 | 4.1285 ± 1.9E-03 | 50 ± 5 | 161 ± 3.4 | 14.63 ± 0.03 | 60.22 ± 0.2 |
| 11 | 3.5053 ± 1.7E-03 | 50 ± 5 | 161 ± 3.4 | 14.63 ± 0.03 | 64.01 ± 0.2 |
| 12 | 3.2045 ± 9E-04 | 50 ± 5 | 161 ± 3.4 | 14.63 ± 0.03 | 66.11 ± 0.3 |
| **Exp.H** | | | | | |
| 1 | 1.9857 ± 8E-04 | 50 ± 5 | 164 ± 3.4 | 14.77 ± 0.03 | 67.89 ± 2.5 |
| 2 | 1.9872 ± 1.0E-03 | 50 ± 5 | 261 ± 5.9 | 14.77 ± 0.03 | 88.08 ± 1.3 |
| 3 | 1.9955 ± 1.0E-03 | 50 ± 5 | 348 ± 8.3 | 14.77 ± 0.03 | 92.0 ± 5.1 |
| 4 | 1.9995 ± 8E-04 | 50 ± 5 | 357 ± 8.9 | 17.36 ± 0.03 | 93.9 ± 5.0 |
| 5 | 2.0099 ± 8E-04 | 50 ± 5 | 342 ± 8.3 | 19.94 ± 0.03 | 96.1 ± 4.0 |
| 6** | 2.0021 ± 2E-03 | 50 ± 5 | 342 ± 8.2 | 22.80 ± 0.03 | 98.99 ± 0.1 |
| 7** | 2.0046 ± 3E-03 | 50 ± 5 | 265 ± 25 | 22.80 ± 0.03 | 96.74 ± 0.3 |
| 8** | 2.0061 ± 4E-03 | 50 ± 5 | 173 ± 20 | 22.80 ± 0.03 | 87.33 ± 0.1 |
| 9** | 2.0076 ± 6E-03 | 50 ± 5 | 128 ± 10 | 22.80 ± 0.03 | 77.30 ± 0.1 |
| **Exp.I** | | | | | |
| 1 | 2.0471 ± 7E-04 | 50 ± 5 | 164 ± 3.4 | 7.92 ± 0.03 | 46.09 ± 1.8 |
| 2 | 2.0565 ± 9E-04 | 50 ± 5 | 164 ± 3.5 | 10.13 ± 0.03 | 56.59 ± 0.2 |
| 3 | 2.0586 ± 1.0E-03 | 50 ± 5 | 163 ± 3.5 | 12.54 ± 0.03 | 64.29 ± 0.1 |
| 4 | 2.0606 ± 1.1E-03 | 50 ± 5 | 163 ± 3.6 | 15.14 ± 0.03 | 70.31 ± 0.1 |
| 5 | 2.0627 ± 1.1E-03 | 50 ± 5 | 164 ± 3.6 | 17.71 ± 0.03 | 75.09 ± 0.1 |
| 6 | 2.0690 ± 1.4E-03 | 50 ± 5 | 162 ± 3.6 | 20.63 ± 0.03 | 80.28 ± 0.07 |
| 7 | 2.0710 ± 1.5E-03 | 50 ± 5 | 161 ± 3.5 | 23.63 ± 0.03 | 83.25 ± 0.04 |





**Table D4.** Experimental data for the N$_2$O experiments using the G5131-i for N$_2$O analysis. Columns: Experimental steps, initial [CH$_4$] in ppm, residence time in seconds, removal efficiency in %, [N$_2$O] in ppb, d$^{15}N\alpha$, d$^{15}N\beta$ and d$^{18}O$ refeer to the three isotypes of N$_2$O. Each of the three isotope values have been corrected for the effects of oxygen, CO and N$_2$O variation according to the method described in Harris et al. (2020). The values have not been bound to an absolute scale by the use of calibration gas, so the daily isotopes levels unaffected by methane are shown in the day.

| Experiment (#) | CH$_{4\,initial}$ (ppm) | $t_R$ (s) | R.E. (%) | N$_2$O (ppb) | d$^{15}N\alpha$ (‰) | d$^{15}N\beta$ (‰) | d$^{18}O$ (‰) |
|---|---|---|---|---|---|---|---|
| Exp.J | | | | | | | |
| Corrected values | | | | | -25.8 ±1.0 | 8.0 ± 1.1 | 106.4 ± 1.4 |
| 1 | 2.4048 ± 6E-03 | 64 ± 5 | 28.3 ± 0.5 | 340.2 ± 0.9 | -22.6 ± 6.3 | 6.3 ± 7.7 | 109.0 ± 4.9 |
| 2 | 2.4048 ± 6E-03 | 64 ± 6 | 29.5 ± 0.2 | 338.3 ± 0.3 | -21.3 ± 11 | 8.8 ± 8.7 | 110.1 ± 5.0 |
| 3 | 2.4048 ± 6E-03 | 86 ± 7 | 34.2 ± 0.2 | 339.5 ± 0.5 | -22.2 ± 10 | 8.0 ± 10 | 109.0 ± 5.6 |
| 4 | 2.4048 ± 6E-03 | 128 ± 10 | 52.2 ± 0.1 | 338.2 ± 0.3 | -24.1 ± 7.3 | 7.0 ± 8.6 | 108.4 ± 5.0 |
| 5 | 2.4048 ± 6E-03 | 513 ± 40 | 84.8 ± 0.1 | 354.9 ± 0.5 | -24.8 ± 7.2 | 8.4 ± 9.0 | 105.6 ± 5.0 |
| Exp.K | | | | | | | |
| Corrected values | | | | | -25.2 ± 0.8 | 9.9 ± 1.5 | 105.2 ± 1.0 |
| 1 | 2.419 ± 1.0E-02 | 117 ± 9 | 37.4 ± 2.7 | 342.5 ± 0.5 | -20.7 ± 6.2 | 8.0 ± 8.2 | 107.3 ± 4.9 |
| 2 | 2.430 ± 2E-03 | 117 ± 9 | 44.2 ± 0.3 | 337.2 ± 0.3 | -22.9 ± 8.2 | 10.0 ± 7.9 | 107.3 ± 5.4 |
| Exp.L | | | | | | | |
| Corrected values | | | | | -40.2 ± 0.7 | 17.9 ± 1.7 | 100.1 ± 0.5 |
| 1 | 2.268 ± 1E-03 | 117 ± 9 | 43.5 ± 2.0 | 316.4 ± 0.5 | -36.5 ± 5.3 | 16.2 ± 5.8 | 101.7 ± 4.1 |
| 2 | 2.406 ± 4.0E-02 | 89 ± 7 | 38.0 ± 1.3 | 329.8 ± 0.9 | -36.8 ± 12.8 | 18.4 ± 8.6 | 102.3 ± 8.9 |
| 3 | 2.406 ± 3E-02 | 135± 10 | 54.3 ± 6.8 | 337.8 ± 2.0 | -37.6 ± 5.5 | 17.9 ± 5.6 | 101.6 ± 3.7 |
| 4 | 2.4018 ± 3E-03 | 86 ± 7 | 37.3 ± 0.8 | 337.7 ± 1.5 | -36.4 ± 14.8 | 19.5 ± 9.1 | 101.8 ± 7.7 |
| 5 | 2.4018 ± 7E-03 | 141 ± 11 | 56.8 ± 0.5 | 338.2 ± 1.3 | -38.5 ± 5.7 | 17.6 ± 6.2 | 101.2 ± 4.0 |

**Table D5.** Table summarizing experiments and setups.

| Setup | Experiment |
|---|---|
| HPXL | A |
| STH-MFC-PD | B |
| STH-PD | C, D, E |
| MTH-PD | F, G, H, I |

## D1.1 HPXL setup

The xenon-lamp experiments shown in Figure D1a were performed to confirm that the Cl$_2$ added to the gas-mix could make it to the photolysis-chamber. The RE% of methane was found as a result of varying the [Cl$_2$] to 16.7, 25, 50, and 92 ppm as seen





**Figure D1.** Exp. A-I is shown. Each illuminated step has been highlighted. **a.** The $CH_4$ is seen as a function of time. The $[Cl_2]$ is varied.**b.** The light intensity and $[Cl_2]$ are varied. **c.** Steps 7-10 highlighted. The light intensity and $t_R$ are varied. **d.** Steps 1-10 are highlighted.The light intensity and $t_R$ are varied. **e.** Steps 1-16 are highlighted. The light intensity and $[Cl_2]$ are varied. Following the initial illumination at 13 W the sample was illuminated at three different $p_{IN}$ for five different chlorine concentrations. The $p_{IN}$ was in the order 13W, 10W and 17W with chlorine steps 20, 40, 50, 60 and 70 ppm. **f.** Steps 1-7 are highlighted. The light intensity are varied. **g.** Steps 1-12 are highlighted. The $CH_4$ level is varied, while the light intensity is kept the same. *h.* Steps 1-9 are highlighted. The light intensity and $t_R$ are varied. **i.** Steps 1-7 are highlighted. The light intensity are varied. Prolonged and stable photolysis enabled due to cooling. Increasing levels of $p_{IN}$ for the photochemical chamber defines the seven different steps.

in Figure D2g. Each concentration step was given 10 minutes to stabilize before the xenon-lamp was turned on for ten minutes. The gas provided to the system was a dynamic mix of flows from three different flasks (see table Gas flasks Table B1). Due to this, it was possible to vary the abundance of chlorine while keeping $[CH_4]$ constant. The experiment confirmed that the level



**Figure D2. a.** RE% as a function of $p_{IN}$ for exp. B1-B4. **b.** Experiment steps D2 and D6-D10: RE% as a function of $p_{IN}$ (W). **c.** RE% of methane plotted against $p_{IN}$ in W. The result from three experiments, D (Square), F (Circle) and I (Arrow) have different settings in $t_R$, [CH$_4$] and [Cl$_2$]. **d.** Experimental steps D1-D5: $t_R$ (s) in the photochemical device as a function of $t_R$ in seconds. **e.** The resulting removal efficiencies of exp. H plotted against $t_R$. An additional zoom inset Figure on the four points around 350 s reveals the removal effect plotted against power. **f.** RE% of methane plotted against $t_R$ in s. The result from three experiments, C (Black), D (Green) and H (Purple) have different settings in $p_{IN}$, [CH$_4$] and [Cl$_2$] **g.** RE% as a function of [Cl$_2$] in ppm for Xenon lamp exp. A. **h.** Resulting RE% plotted against [Cl$_2$] in ppm for exp. steps E2-E16. Three different power settings are used. 9.9 W (Diamond), 13.4 W (Circle) and 17.5 W (Square). **i.** The RE% is displayed as a function of the initial methane concentration with the remaining fixed parameters such as Cl$_2$ mixing ratio, $t_R$, and $p_{IN}$. The three points (Star) in the Figure represent steps suffering from early- experiments-deviation.

of Cl$_2$ could be controlled and that higher levels resulted in greater depletion of methane.





## D1.2 STH-MFC-PD setup

In exp. B, $P_{IN}$ was varied in steps one to four, presented in Figure D1b. The aim was to determine the effect of varying light intensity. Figure D2a shows the RE% as a function of $p_{IN}$ for experiments one to four. The initial methane concentration is maintained at $3.68 \pm 0.02$ ppm. Step one and two are both examples of the start-up-deviation. At the time of step three and four, sufficient flushing had taken place. The chlorine concentration was increased from 16.7 to 50 ppm starting with step 5.

**Table D6.** exp. B. The three experimental steps clearly shows an increasing RE% as the $p_{IN}$ and the $Cl_2$ mixing ratio are increased

| Step | $CH_4$ | $Cl_2$ | Residence time | Power | RE% |
|------|--------|--------|----------------|-------|-----|
| (#) | (ppm) | (ppm) | (s) | (W) | (%) |
| B3 | $3.7069 \pm 1.1 \cdot 10^{-4}$ | $16.7 \pm 1.5$ | $62.2 \pm 1.5$ | $9.91 \pm 0.03$ | $6.87 \pm 0.01$ |
| B5 | $3.919 \pm 1.4 \cdot 10^{-3}$ | $50 \pm 5$ | $61.0 \pm 1.4$ | $9.92 \pm 0.03$ | $23.91 \pm 0.05$ |
| B6 | $3.945 \pm 1.3 \cdot 10^{-3}$ | $50 \pm 5$ | $61.4 \pm 1.4$ | $16.59 \pm 0.03$ | $32.69 \pm 0.04$ |

The four relevant variables and resulting RE% can be seen in Table D6. $[Cl_2]$ was increased by a factor of 2.5 between steps 3 and 5. The increase results in a 3.5 fold increase of RE%. Furthermore, the $p_{IN}$ is increased when going from step 5 to 6 which also leads to an increase in RE%. The comparison between these three steps, the positive relation for both chlorine concentration and $p_{IN}$ on the RE% was confirmed.

## D1.3 STH-PD setup

Four experiments (C, D and E) used this setup. Exp. C presented in Figure D1c was carried out with constant supply of $[Cl_2]$ at 50 ppm and $[CH_4]$ at $3.981 \pm 0.018$ ppm. Step two and three had the same $t_R$, as does step one and four. In addition, the experiments vary in $p_{IN}$ as can be seen in Table D7. Table D7 shows how the combination of increased $t_R$ and $p_{IN}$ yields a higher RE% . The exp. D was carried out with $[Cl_2]$ kept constant at 32 ppm. The initial methane concentration was maintained

**Table D7.** Exp. C

| Experiment | $CH_4$ | $Cl_2$ | Residence time | Power | RE% |
|------------|--------|--------|----------------|-------|-----|
| (#) | (ppm) | (ppm) | (s) | (W) | (%) |
| 1 | $3.957 \pm 5 \cdot 10^{-3}$ | $50 \pm 5$ | $41.4 \pm 1.0$ | $9.63 \pm 0.03$ | $18.90 \pm 0.11$ |
| 2 | $4.0301 \pm 5 \cdot 10^{-4}$ | $50 \pm 5$ | $41.4 \pm 1.0$ | $17.30 \pm 0.03$ | $27.83 \pm 0.06$ |
| 3 | $3.955 \pm 2 \cdot 10^{-3}$ | $50 \pm 5$ | $129 \pm 3$ | $9.63 \pm 0.03$ | $44.51 \pm 0.3$ |
| 4 | $3.986 \pm 9 \cdot 10^{-4}$ | $50 \pm 5$ | $128 \pm 3$ | $17.30 \pm 0.03$ | $60.6 \pm 0.5$ |

at $3.547 \pm 0.005$ ppm. Similarly to exp. C the $t_R$ and $p_{IN}$ were varied. Steps one to five are carried out with the same $p_{IN}$ in the device but with varying residence times, see Figure D2f and D2d. In Figure D2f the data for exp. D exhibits a clear agreement between $t_R$ and RE% . The longer $t_R$ within the photochamber results in greater removal efficiencies. Steps two





and six to ten are carried out with the same $t_R$ but with varying $p_{IN}$, see Figure D2c and D2b.

The experimental steps of exp. E, Figure D1e, were held the same initial methane concentration at $3.39 \pm 0.01$ ppm and

the same $t_R$ at $60.82 \pm 0.18$ s. Throughout the experiments, three levels of $p_{IN}$ were tested against varied levels of $Cl_2$ mixing ratio spanning in the range 20 - 70 ppm. Figure D2h presents, looking at 20 ppm $Cl_2$, that a greater $p_{IN}$ yields higher RE%.

### D1.4   MTH-PD setup

Four experiments (F, G, H and I) were done with this setup. The exp. F, Figure D1f, was run at a constant level of $[CH_4]_{inititial}$

at $3.593 \pm 0.019$ ppm and $[Cl_2]$ at $50 \pm 5$ ppm. At a flow kept at 15.5 ml min$^{-1}$ the $t_R$ in the photochamber was maintained at $161.06 \pm 3$ s. Across exp. F the step-wise changes were made for $p_{IN}$ ranging from 6.75-22.92 W. The daily measurement is presented in Figure D1f; where the removal for the steps, with the exception of the first step, is characterized by an initial RE% but this efficiency drops during the first five minutes of illumination. The relationship found between removal and $p_{IN}$ for exp. F can be seen in Figure D2c.

Exp. G was carried out with a step-wise change of $[CH_4]_{inititial}$ in the range 1.39 to 4.13 ppm, at constant $t_R$ of 164 s, $[Cl_2]$ at 50 ppm, and $p_{IN}$ 14.6 W. The daily result can be seen in Figure D1g, where the improvement of silicone removal can be observed from stable levels of RE%. As can be seen in Figure D2i decreasing the initial methane concentration yields, as expected, a greater RE% .

Exp.H was carried out with the constant $[CH_4]_{inititial}$ at $2.000 \pm 0.003$ ppm and $Cl_2$ mixing ratio at $50 \pm 5$ppm, but with

mixed settings of $t_R$ and power. Step S1-S3 were done with constant power at 14.8 W with $t_R$ increasing from 164-350 s. Then keeping $t_R$ around 350 s three steps of increasing power were tested, ranging from 14.8-22.8 W. In between step S4 and S5 a fan was installed. The final three step were kept at 22.8 W and stepped through reduced $t_R$ from 342-130 s.

Exp. I was carried out with $[CH_4]_{inititial}$ maintained around $2.01 \pm 0.01$ppm, $[Cl_2]$ at 50 ppm and the $t_R$ held at $163.1 \pm 0.4$ s. The only parameter varied was the $p_{IN}$ to the photochemical device. The light was turned at 7.9 W and was left on for the

duration of the experiments with a step-wise increase in $P_{IN}$ after stable removal had been maintained for 5 min. The resulting methane concentration can be seen in Figure D1i. $[CH_4]$ increases throughout the experiments due to the chlorine-pressure-decline. For the purpose of calculating RE% , the expected $[CH_4]$ for each of the steps was fitted from the initial $[CH_4]$ and the end $[CH_4]$; $CH_4 = 0.0002 \cdot t + 2.0461$. The relative median values of initial methane and $t_R$ were chosen in order to best resolve the effects of varying $p_{IN}$. As the removal effect approaches 100 % asymptotically, the sensitivity to changes will be

greater at lower removal values.

The results presented for exp. I in Figure D2c can be compared to the results from exp. D and F and represents the improvements implemented to the system. Unlike for those experiments, the trend of exp. I is explained by one trend asymptotically approaching 100 % removal.





### D1.5 Comparison

Figure D2c shows a comparison of three different experiments where $p_{IN}$ was varied. When comparing experiments F and I the improvement in performance of the device is clear. However, even if $t_R$ and $[Cl_2]$ are identical, the initial methane concentration of exp. F is 3.59 ppm compared to exp. I at 2.096 ppm. Exp. D alone shares some $P_{IN}$ levels and is operated at the same initial methane level as exp. F. The $t_R$ and $[Cl_2]$ are lower and a lesser removal is accordingly expected. Hence, the main thing to observe is behavior at higher $P_{IN}$. The efficiency of the photochamber decreases as seen in exp. D and F.

The improvements done on the photochamber and installation of a fan to cold the photochemical chamber have prolonged the lifetime of the chamber and improved efficiency.

Figure D2f shows a comparison of three different experiments where $t_R$ was varied and in some cases $P_{IN}$ as well. $t_R$ is improved in the manner that the MTH-PD setup made it possible to obtain higher $t_R$ and more efficient use of the photochemical chamber. The experiments with a single tube do not have long residence times. As seen in Figure D2f longer $t_R$ greatly

improves the RE% and is, therefore, essential to further improve the setup.





## D2 N₂O experimental results

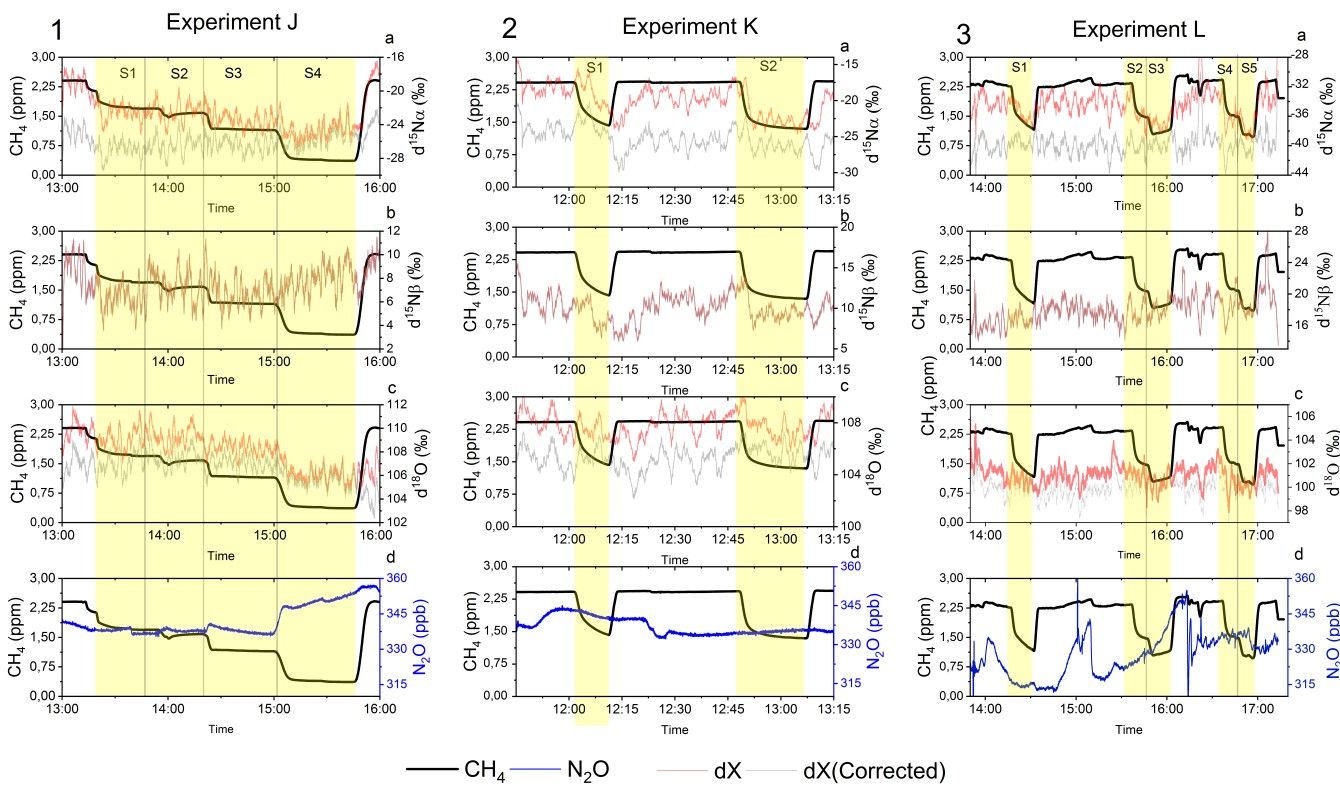

**Figure D3.** Results from the three experiments J, K and L using the G5131-i for N₂O isotope measurements. CH₄ level is depicted in each figure in ppm along the first y-axis. Highlights indicate the several different oxidation settings. **Row 1** Measurements of d$^{15}$N$\alpha$ in ‰ plotted along the second y-axis. Red highlights a 100 s averaged measured values corrected for O₂, CO and CO₂ effects, while grey indicates a 100 s average value that have been corrected for all interference including CH₄. **Row 2** Measurements of d$^{15}$N$\beta$ in ‰ plotted along the second y-axis. Red highlights a 100 s averaged measured values corrected for O₂, CO and CO₂ effects, while grey indicates a 100 s average value that have been corrected for all interference including CH₄. **Row 3** Measurements of d$^{18}$O in ‰ plotted along the second y-axis. Red highlights a 100 s averaged measured values corrected for O₂, CO and CO₂ effects, while grey indicates a 100 s average value that have been corrected for all trace-gas interference including CH₄. **Row 4** Measurements of [N₂O] in ppb shown in blue. Variation observed correspond to fluctuations in the mixing of the three gasses. **Exp. J** In this experiment the light was turned on throughout the entire experiment, with the experimental steps corresponding to changes in $t_R$. **Exp. K** In this experiment two experimental steps were used with different power settings. **Exp. L** In this experiment the a sofnocat trap was used in the first three experimental steps, while 4 and 5 were completed without. The variation between experimental steps correspond to changes in $t_R$.





From the experiment investigating the compatibility of the removal method and the analysis of $N_2O$, it was found that the oxidization had no effect on the $N_2O$ abundance nor isotopic composition. It was however discovered that the oxidation path for $CH_4$ terminated at CO, as the isotopic signal changed matching the interference of CO. To remove this effect a sofnocat

trap was implemented, which oxidize the CO to $CO_2$. By applying the tracegas and matrix corrections described in Harris et al. (2020), it was found that the isotopic levels remained stable across the oxidation. Variation observed in the $N_2O$ was due to the unstable supply of $Cl_2$, resulting in slight shifts in the dilution. The value of $d^{15}N\alpha$ and $d^{18}O$ were both found to approach the unaffected target value during the oxidation as was hoped. Results are shown in Figure D3



## Appendix E: Kintecus reactions and results

The results from the kinetic model are shown in Figure E1.

**Table E1.** JPL: Burkholder et al. (2020).* Third order rate expression in the units ($cm^6$ molecules$^{-2}$ s$^{-1}$). Hossaini: Hossaini et al. (2016)

| Reaction | Reaction Rate Coefficient ($cm^3$ molecules$^{-1}$ s$^{-1}$) | Reference |
|---|---|---|
| $Cl_2 \rightarrow 2\ Cl$ | X | |
| $Cl + Cl + M \rightarrow Cl_2 + M$ | 1.29E-32* | Baulch et al. (1981) |
| $O_2 + CH_3 \rightarrow CH_3O_2$ | 1.79E-12 | Atkinson et al. (1989) |
| $O_2 + CH_3O \rightarrow CH_2O + HO_2$ | 1.65E-15 | Orlando et al. (2003) |
| $O_2 + HCO \rightarrow CO + HO_2$ | 5.20E-12 | Atkinson et al. (2001) |
| $O_2 + CH_2Cl \rightarrow CH_2ClO_2$ | 2.91E-12 | JPL |
| $Cl + CH_3O \rightarrow CH_2O + HCl$ | 1.91E-11 | Daële et al. (1996) |
| $Cl + CH_3OH \rightarrow CH_3O + HCl$ | 5.50E-11 | JPL |
| $Cl + CH_2O \rightarrow HCO + HCl$ | 7.32E-11 | JPL |
| $Cl + Cl_2O \rightarrow Cl_2 + ClO$ | 9.60E-11 | JPL |
| $Cl + CH_3Cl \rightarrow CH_2Cl + HCl$ | 4.98E-13 | JPL |
| $Cl + CH_2Cl_2 \rightarrow CHCl_2 + HCl$ | 3.57E-13 | JPL |
| $Cl + CCl_3 \rightarrow CCl_4$ | 6.51E-11 | Ellermann (1992) |
| $Cl + CHCl_3 \rightarrow HCl + CCl_3$ | 1.20E-13 | JPL |
| $Cl + CH_3O_2 \rightarrow CH_3O + ClO$ | 1.60E-10 | JPL |
| $Cl + CH_3O_2 \rightarrow CH_2O_2 + HCl$ | 1.60E-10 | JPL |
| $Cl + CH_4 \rightarrow CH_3 + HCl$ | 1.07E-13 | Bryukov et al. (2002) |
| $Cl + CHClO \rightarrow HCl + Cl + CO$ | 7.79E-13 | Atkinson et al. (2001) |
| $Cl + H_2O_2 \rightarrow HCl + HO_2$ | 4.10E-13 | JPL |
| $Cl + CH_3 \rightarrow CH_3Cl$ | 1.61E-12 | Kaiser (1993) |
| $Cl_2 + CH_2Cl \rightarrow CH_2Cl_2 + Cl$ | 2.54E-13 | Seetula (1998) |
| $Cl_2 + CHCl_2 \rightarrow CHCl_3 + Cl$ | 2.25E-14 | Seetula (1998) |
| $Cl_2 + CH_3 \rightarrow CH_3Cl + Cl$ | 1.55E-12 | Eskola et al. (2008) |
| $Cl_2 + HCO \rightarrow CHClO + Cl$ | 5.59E-12 | Timonen et al. (1988) |
| $Cl_2 + OH \rightarrow HClO + Cl$ | 6.42E-14 | Atkinson et al. (2007) |





**Table E2.** JPL: Burkholder et al. (2020).* Third order rate expression in the units ($cm^6$ molecules$^{-2}$ s$^{-1}$). Hossaini: Hossaini et al. (2016)

| Reaction | Reaction Rate Coefficient ($cm^3$ molecules$^{-1}$ s$^{-1}$) | Reference |
|---|---|---|
| $OH + CH_4 \rightarrow CH_3 + H_2O$ | 6.30E-15 | Bonard et al. (2002) |
| $OH + CH_3OOH \rightarrow H_2O + CH_3O_2$ | 7.40E-12 | JPL |
| $OH + CH_3OOH \rightarrow CH_2O + OH + H_2O$ | 7.40E-12 | JPL |
| $OH + CH_2O \rightarrow HCO + H_2O$ | 8.50E-12 | JPL |
| $OH + HCl \rightarrow Cl + H_2O$ | 7.80E-13 | JPL |
| $OH + HClO \rightarrow ClO + H_2O$ | 5.00E-13 | Atkinson et al. (2007) |
| $OH + CH_2Cl_2 \rightarrow CHCl_2 + H_2O$ | 1.00E-13 | JPL |
| $OH + CHCl_3 \rightarrow CCl_3 + H_2O$ | 1.00E-13 | JPL |
| $OH + CH_3O \rightarrow CH_2O + H_2O$ | 3.01E-11 | JPL |
| $OH + CH_3OH \rightarrow CH_3O + H_2O$ | 1.40E-13 | Atkinson et al. (2001) |
| $OH + CH_3 \rightarrow CH_3OH$ | 9.30E-11 | Oser et al. (1992) |
| $OH + CH_2ClOOH \rightarrow CH_2ClO_2 + H_2O$ | 3.60E-12 | Hossaini |
| $OH + CH_2ClOH \rightarrow CH_3O + HClO$ | 4.54E-14 | Hossaini |
| $OH + H_2O_2 \rightarrow HO_2 + H_2O$ | 1.80E-12 | JPL |
| $OH + CHClO \rightarrow Cl + CO + H_2O$ | 3.20E-13 | Hossaini |
| $OH + ClO \rightarrow Cl + HO_2$ | 1.80E-11 | JPL |
| $OH + ClO \rightarrow HCl + O_2$ | 1.30E-12 | JPL |





**Table E3.** JPL: Burkholder et al. (2020).* Third order rate expression in the units ($cm^6$ molecules$^{-2}$ s$^{-1}$).** First order rate expression in the units s$^{-1}$ Hossaini: Hossaini et al. (2016)

| Reaction | Reaction Rate Coefficient ($cm^3$ molecules$^{-1}$ s$^{-1}$) | Reference |
|---|---|---|
| $HO_2 + CH_3O_2 \rightarrow CH_3OOH + O_2$ | 5.12E-12 | JPL |
| $HO_2 + Cl \rightarrow HCl + O_2$ | 3.50E-11 | JPL |
| $HO_2 + Cl \rightarrow ClO + OH$ | 9.30E-12 | JPL |
| $HO_2 + ClO \rightarrow HClO + O_2$ | 6.90E-12 | JPL |
| $HO_2 + CH_3O \rightarrow CH_2O + H_2O_2$ | 5.00E-13 | Tsang and Hampson (1986) |
| $HO_2 + HO_2 \rightarrow H_2O_2 + O_2$ | 1.60E-12 | Atkinson et al. (2004) |
| $HO_2 + CH_2ClO_2 \rightarrow CH_2ClOOH + O_2$ | 5.01E-12 | Hossaini |
| $HO_2 + CH_2ClO_2 \rightarrow CHClO + H_2O + O_2$ | 5.01E-12 | Hossaini |
| $ClO + ClO \rightarrow O_2 + Cl_2$ | 4.91E-15 | JPL |
| $ClO + ClO \rightarrow 2Cl + O_2$ | 8.00E-15 | JPL |
| $ClO + Cl + M \rightarrow Cl_2O + M$ | 1.56E-32* | Xu (2010) |
| $ClO + CH_3O_2 \rightarrow Cl + O_2 + CH_3O$ | 2.40E-12 | JPL |
| $ClO + CH_3 \rightarrow CH_3OCl$ | 5.69E-11 | Brudnik et al. (2009) |
| $CH_3O_2 + CH_3O_2 \rightarrow CH_3O + CH_3O + O_2$ | 3.50E-13 | JPL |
| $CH_3O_2 + CH_3O_2 \rightarrow CH_3OH + CH_2O + O_2$ | 3.50E-13 | JPL |
| $CH_3 + CH_3O_2 \rightarrow CH_3O + CH_3O$ | 4.50E-11 | Pilling and Smith (1985) |
| $CH_3O + CH_3O \rightarrow CH_2O + CH_3OH$ | 3.85E-11 | Hassinen and Koskikallio (1979) |
| $CH_2ClO_2 + CH_3O_2 \rightarrow CH_2ClO + CH_2O + HO_2$ | 2.50E-12 | Hossaini |
| $CH_2ClO_2 + CH_3O_2 \rightarrow CH_2ClOH + CH_2O + O_2$ | 2.50E-12 | Hossaini |
| $CH_2ClO_2 + CH_3O_2 \rightarrow CHClO + CH_3OH + O_2$ | 2.50E-12 | Hossaini |
| $CH_2ClO_2 + CH_2ClO_2 \rightarrow CH_2ClO + CH_2ClO + O_2$ | 3.50E-12 | Hossaini |
| $CH_2Cl_2 + Cl \rightarrow CHCl_2 + HCl$ | 3.57E-13 | Atkinson et al. (2001) |
| $CHCl_2 + Cl_2 \rightarrow CHCl_3 + Cl$ | 2.25E-14 | Seetula (1998) |
| $CCl_3 + Cl \rightarrow CCl_4$ | 6.51E-11 | Ellermann (1992) |
| $HCO + Cl_2 \rightarrow HC(O)Cl + Cl$ | 5.59E-12 | Timonen et al. (1988) |
| $CH_2O_2 \rightarrow CO + H_2O$ | 6.00E+04** | Maricq et al. (1994) |

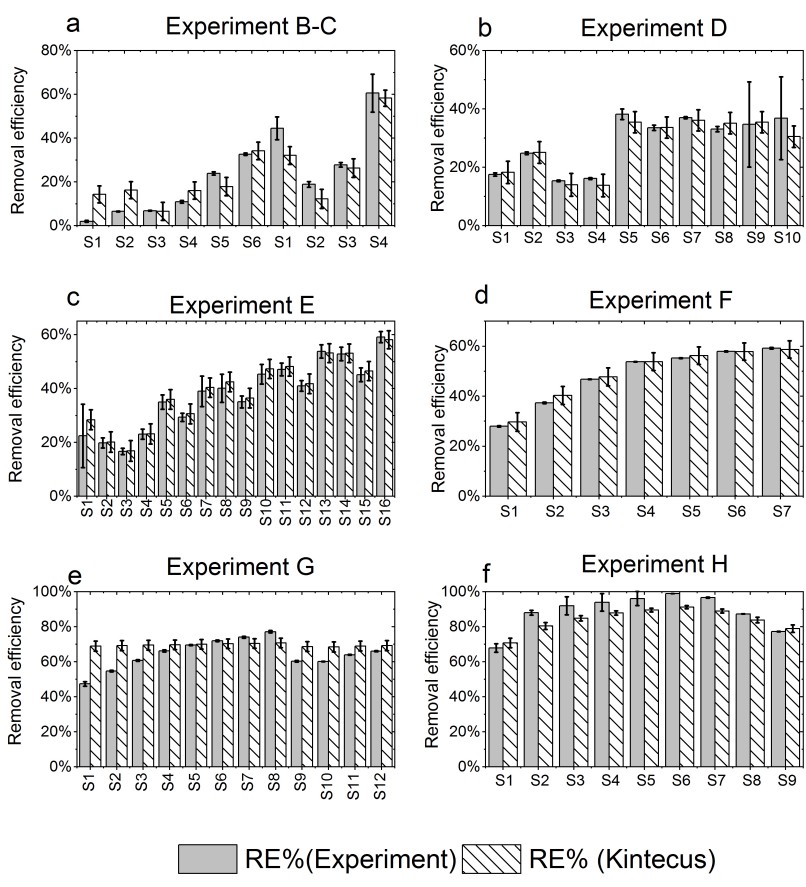

**Figure E1.** RE% as found experimentally (Grey) and by model ( White stripes). **a.** Exp.B and C **b.** Exp.D **c.** Exp.E **d.** Exp.F **e.** Exp.G **f.** Exp.H



*Author contributions.* Matthew S. Johnson, Merve Polat and Jesper Baldtzer Liisberg conceived and planned the experiments. Merve Polat and Jesper Baldtzer Liisberg carried out the experiments. Merve Polat, Jepser Baldtzer and Morten Krogsbøll planned and carried out the simulations. Merve Polat and Jesper Baldtzer Liisberg contributed to the interpretation of the results. Merve Polat and Jesper Baldtzer Liisberg wrote the manuscript in consultation with Thomas Blunier and Matthew S. Johnson. All authors provided critical feedback and

575 helped shape the research, analysis and manuscript.

*Competing interests.* The authors declare that they have no conflict of interest.

*Data availability.* All data are available from the corresponding author upon request.

*Acknowledgements.* We thank the Copenhagen Center for Atmospheric Research (CCAR), the Centre for Ice and Climate and the University of Copenhagen.





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
