# Peer review of "Photochemical method for removing methane interference for improved gas analysis"

_Atmospheric Measurement Techniques, 2021_

## Referee Comment (RC1)

**General comments:**

The manuscript by Merve Polat et al. describes the development and optimization of novel method for $CH_4$ removal from air via chlorine-initiated oxidation with the focus to minimize spectral interferences during $N_2O$ isotope analysis by CRDS (Picarro G5131-i). The study includes the design and validation of a proof-of-concept device and the validation of a kinetic model to predict the dependence of the $CH_4$ removal efficiency on methane concentration, chlorine photolysis rate, chlorine concentration, and residence time.

I find the manuscript timely as the strong $CH_4$ interference of the Picarro $N_2O$ isotope analyzer was recently identified by Harris et al. (AMT, 2020). Therefore, a technique for specific $CH_4$ removal is of high interest for the users of these analyzers but complicated. An alternative concept would be removal of N2O and release in a well-defined gas matrix as applied by IRMS and also laser spectroscopy.

I doubt whether addition of a toxic substance such as $Cl_2$ should be propagated for $CH_4$ removal. I therefore assume (hope) the study is more focusing on the feasibility of such a technique than suggesting its use.

The wording of the manuscript should be strongly improved by careful proofreading! In addition the technical quality is not yet good enough, spaces are missing, brackets are not closed, superscripts are not used, etc., please carefully check for this.

In addition the structure of the experimental section should be streamlined to improve readability (see below).

Further, I have a number of specific comments and technical corrections the authors should consider before publication is feasible.

**Specific comments:**

The abbreviations applied for setups, experiments and flasks in Table 1 +2 are confusing. Why not using one simplified setup (Figure 1), which guides the reader in a better way (see specific comments below).

The experiments in the results section should be streamlined, only experiments required to tell the main story should be selected, ordered in a well-motivated manner, and main results provided in the main text of the manuscript. Currently all results are mentioned in the results section but removal efficiencies only given in the appendix (D1 – D3).

The modelling and experimental results should be combined to cross-validate each other. Why not displaying the modelled functions in e.g. Figure 4?

**Technical corrections:**

**Page 1 Line 19:** Possibly "High-resolution instruments targeting at specific rovibrational transitions ..." or similar.

**Page 2 Line 27:** Please change to "$\delta^{15}N^{\alpha}$, $\delta^{15}N^{\beta}$", here and elsewhere in the text.

**Page 2 Lines 27 – 28:** The term "biological activity in agricultural soils" is very unspecific; in addition the message of the sentence is unclear. Would the following sentence fit better? "These instruments are often used to measure isotopic signatures of $N_2O$ emitted from soils (Ibraim et al. (2019a), Wolf et al. (2015)), which can help to differentiate different microbial and abiotic production pathways."

**Page 2 Lines 29 – 30:** The subsequent sentence might be changed to: "$N_2O$ formation in soils is commonly accompanied by production / uptake of other trace gases such as $CH_4$, $CO_2$, and water vapor (Erler et al. (2019), Ibraim et al. (2019b))."

**Page 2 Line 30 – 31**: The message of the sentence "In some samples, changes in $CH_4$ and $CO_2$ can exceed 1.8 ppm and 200 ppm, respectively (M. Zimnoch and Rozanski (2010))." is unclear and the numbers seem arbitrary. Please provide a more profound statement on $N_2O$, $CH_4$, $CO_2$ concentrations above different soils. I would suggest to give a high and low emission scenario for both $CH_4$ and $CO_2$. This should give the reader a feeling on usual $CH_4$ and $CO_2$ changes along with targeted $N_2O$ concentration (and isotope) changes.

**Page 2 Line 31 – 33**: A number "5 ‰" is provided without a statement on whether the main interference is from $CH_4$ or $CO_2$ and which delta values are affected most. Please give the deviations for a specific delta value for the high and low $CH_4$, $CO_2$ emission scenario (statement above).

**Page 2 Line 33:** The statement "improve the accuracy by controlling these interferences" is questionable? Possibly "Improve the accuracy by either removing the interfering trace gases (i.e. $CH_4$, $CO_2$) or the target substance ($N_2O$) or the analysis of interferants and implementation of a correction algorithm."

**Page 2 Line 35:** How should the "a careful determination of the calibration curve.(Kantnerová et al. (2020))" help? Please rephrase or delete this sub-sentence.

**Page 2 Line 38:** Place replace "continued" by "on-line" or "monitoring applications" or similar.

**Page 3 Table 1 and 2:** The wording of setups should be simplified to support the readers, e.g. using "Xe lamp" instead of "XPXL", or "Photochemical Device" instead of "STH-MFC-PD", etc.?

The different setups are quite similar, would it be possible to show one setup only and mention difference in the others?

The experiments are labelled (A-I) same than the flasks (A-D), please change one label.

**Page 3 Line 64:** The end of the sentence "... applying the method to measurements of $N_2O$." should be reformulated.

**Page 3 Line 65 – 66:** The last sentence of the introduction should be rephrased, e.g. "the measured isotopic data of isotopes", "compared to the stable values", etc.

**Page 3 Line 67:** The structure of the method section should be rethought: 2.1.1 Methane experiments; 2.1.2 Post photolysis scrubbing. I assume Post photolysis scrubbing is part of the Methane removal experiments, so should be integrated in 2.1.1?

**Page 3 Line 71:** What is a "Flow-Controlled Chlorine Waste", please rephrase.

**Page 3 Line 70 – 73:** The section should be rephrased considering the final

**Page 4 Figure 1**: Please show one exemplary setup and mention differences for the others?

**Page 4 Line 74 – 82:** First there are some sentences on gas cylinders flows and setup, thereafter a section "Manifold" dealing specifically with the setup? Please combine and rephrase both sections.

**Page 5 Line 98**: The statement "they were found to likewise remove HCl and $Cl_2$" Is unspecific for such an important question. How was this determined and what happens with the Picarro analyzer if it is flushed with HCl or $Cl_2$?

Which chlorinated species pass the Ascarite trap and are scrubbed with the activated carbon trap, please specify? Is it advisable to purge these species through a Picarro CRDS analyzer?

**Page 5 Section 2.1.3**: The authors should mention how measurements were referenced to scales? Were the delta values of the applied $N_2O$ gases known and deviations analyzed for the $CH_4$ addition experiments and the $CH_4$ removal experiments?

**Page 7 section 3.1:** The author show results of experiment H first, but do not indicate how this is motivated? In addition, main results (Table D1 – D3) are hidden in the appendix. I would suggest to streamline the structure of the results section, and provide important results in the main text in a well-motivated order.

**Page 7 Line 151:** "experiment 3 step 5" is this "experiment C step 5"?

**Page 8 Figure 3:** The authors mention in the legend that "the experimental step is indicated at the top" but the experiments are named Sx, while in the text the terms Hx are used, please change?

**Page 8 Line 160 - 162**: The sentence should be reformulated, was it really a pressure drop in the cylinder? If a pressure reduction valve was used this seems unplausible? As a $Cl_2$ sensor was used this should have been noted during experiments

**Page 10 section 3.1.1:** Figure 5 provides the main finding of the manuscript with respect to consistency of delta values for $CH_4$ addition / correction and $CH_4$ removal / correction.

I have some concerns:

1) The authors did not achieve complete $CH_4$ removal, which would be needed to waive the need for $CH_4$ interference correction. What was the reason for this as experimental details for quantitative removal are known from the preceding section?

2) The indicated $\delta^{18}O$ and $\delta^{15}N^{\alpha}$ numbers are highlight unplausible, not normalized to scales? This might be ok for a feasibility study, but why are results not normalized to the $N_2O$ isotopic composition without $CH_4$ addition?

3) Figure 5 shows mainly instrumental noise of the CRDS analyzer, with some plausible drops in the signal, when $CH_4$ was removed.

4) Why did the authors observe variations in $N_2O$ concentrations? This a very critical observation if $N_2O$ was removed by a chemical process and would question the approach. Please comment on this?

---

## Referee Comment (RC2)

Comments on "Photochemical method for removing methane interference for improved gas analysis" by Merve Polat et al.

**General comments:**

This manuscript describes a photochemical scrubbing method, using $Cl_2$, for the removal of $CH_4$ in whole air samples. This described method is intended to enable accurate determination of the isotopic composition of $N_2O$ ($\delta^{15}N^{\alpha}$, $\delta^{15}N^{\beta}$, and $\delta^{18}O$) by minimizing the spectroscopic interference arising from $CH_4$ (among several others), which poses immense challenge. This method indeed shows potential as the removal of $CH_4$ through this photochemical scrubbing does not alter the matrix composition dramatically. In addition to the experimental work, the authors have also complemented their experimental results with predictions using a kinetic model studying all the dependencies. This method in general can be used for any application requiring removal of $CH_4$ (and other hydrocarbons/VOCs), and is not just limited to the measurements involving $N_2O$. And hence is an important development that the scientific community could benefit from.

Although the content of the manuscript is very interesting, consisting of appropriate method-development related experiments and complementing model prediction, the manuscript itself is difficult to follow at times due to: the use of too many abbreviations, and having to go back-and-forth between the main section of the manuscript and the supplementary section where part of the information is. Additionally, there are sections where the texts require rephrasing to make the content more understandable. Please see details in the specific comments section.

While this proof-of-concept method is aimed towards reducing the interferences of $CH_4$ during the measurements of the singly-substituted isotopologues of $N_2O$, the experiments and results shown to demonstrate the applicability is very limited. Since the removal efficiency of $CH_4$ achieved is never ~100%, discussion on its implication was not evaluated thoroughly. This is particularly concerning because one has to then always co-measure methane, which is partially scrubbed, in order to incorporate any possible $CH_4$-dependent correction. So does the method provide any benefit over performing a careful $CH_4$-dependent interference correction? Additional experiments showing the repeatability expected from this method using isotopically-calibrated $N_2O$ samples was also missing.

The manuscript in its current form requires considerable rework and I would recommend publication after all concerns have been addressed.

**Specific comments:**

Line 74 and Table 2: How is tank A calibrated for $[Cl_2]$? Is this a commercial product?

Line 82: The chlorine detector is rated for 0-20 ppm, so how were chlorine concentrations determined in experiments done above 20 ppm, shown later in the manuscript?

Line 85: LED already stands for Light emitting diodes, so should be mentioned "LED" only and not "LED diodes". Please correct this throughout the manuscript.

Line 93: Please provide supplier details/ part number for Krytox™.

Line 96: Magnesium perchlorate is $Mg(ClO_4)_2$.

Line 100: Please provide supplier details, part number, specification for the activated carbon trap used.

Line 103: "*A final set of experiments is conducted using a Picarro CRDS model G5131-i, capable of measuring $N_2O$ mixing ratio and its isotopic abundance.*"

Lines 150-152: Throughout the manuscript, the steps are referred to as S1, S2.... and not as in your example C5 (line 152), please check and revise accordingly to be consistent.

Line 155: "H5": please see my previous comment

Lines 159-162: This paragraph somehow feels very unlinked with the previous paragraph. Please explain the "issue" by pointing the reader to the graph, what exactly to look at? How does the build-up of $H_2O$ happen? Why is the $Cl_2$ raw data not shown along with?

Figure 4: Typically, when you have units shown on the axis label, you don't have to show them on the tick labels, so the % signs on the ticks can be removed. And the abbreviation RE has not been introduced anywhere in the text, so please include this.

Line 187: What is d here? Please define your notation. Is it delta that you are referring to? If so, please describe how often you measure your reference/ calibration etc. Please check and change this throughout the manuscript.

Line 187-188: "*The results are from experiment L, where a sofnocat trap  was installed to remove the CO formed by the $CH_4$ oxidation.*"

Line 189: "…*it was found that the isotopic enrichments* …" Please introduce this to the reader why they should expect isotopic enrichment and not depletion in $\delta^{15}N^{\alpha}$ and $\delta^{18}O$.

Line 190: How stable is the oxidation process in a prolonged time period, e.g. during a continuous 10 hr measurement period, and in practice you would really turn it ON all the time during a measurement. How much of the variability in $[Cl_2]$ translates into your final measurement uncertainty?

Line 193-194: How is the variation in $[N_2O]$ due to variation in $[Cl_2]$, is it not due to dilution? And correct the spelling of variation in Line 194.

Section 3.1.1 (in general): Why was the method with the highest removal efficiency not used here?

Line 229: "… *that an increase in $Cl_2$ concentrations increases the $[CCl_4]$ production (see Figures 7a, 7b and 7d.)*"

Figure 7 (caption): "The Removal efficiency of methane depletion (Black), …" should be "The removal efficiency of methane (Black), …"

Line 233-234: If you use NaOH to remove $CO_2$ from a sample, the matrix changes significantly. To what level of matrix alteration not a problem?

Lines 234-235: "*The $NO_x$ concentration in our experiments is insignificant and hence these reactions have not been included in the model.*"

Line 252: What is the typical concentration range of $Cl_2$ produced by this method? Please elaborate this and describe the calibration and monitoring/ data recording method for $Cl_2$.

Line 285-286: Please rephrase.

Lines 331-336: Please avoid repeating texts already used in the main body of the manuscript (lines 89-93).

Figure B1: Abbreviations are typically introduced once, the first time they come up in the document. So please don't expand your abbreviations every time you describe a figure.

Line 354: How does CO interfere with $N_2O$, please elaborate this and remind the reader which isotopologues are specifically affected.

Figure C1: Why repeat a figure when you can refer to Figure 2?

Lines 407-410: Please rephrase this paragraph and elaborate on "This effect…". The explanation is not clear.

Table D4 (Caption): "… refeer to the three isotypes of $N_2O$." should be something like "… in ‰ refers to the three isotopologue measurements of $N_2O$."

---

## Author Comment (AC1)

General comments:

The manuscript by Merve Polat et al. describes the development and optimization of novel method for $CH_4$ removal from air via chlorine-initiated oxidation with the focus to minimize spectral interferences during $N_2O$ isotope analysis by CRDS (Picarro G5131-i). The study includes the design and validation of a proof-of-concept device and the validation of a kinetic model to predict the dependence of the $CH_4$ removal efficiency on methane concentration, chlorine photolysis rate, chlorine concentration, and residence time.

I find the manuscript timely as the strong $CH_4$ interference of the Picarro $N_2O$ isotope analyzer was recently identified by Harris et al. (AMT, 2020). Therefore, a technique for specific $CH_4$ removal is of high interest for the users of these analyzers but complicated. An alternative concept would be removal of $N_2O$ and release in a well-defined gas matrix as applied by IRMS and also laser spectroscopy.

REPLY. Thanks very much, we are glad you also find it interesting. It would also be possible in theory to do as you suggest, given a suitable separation method, and as long as complications due to the separation are not an issue. Many separation technologies fractionate isotopes.

I doubt whether addition of a toxic substance such as $Cl_2$ should be propagated for $CH_4$ removal. I therefore assume (hope) the study is more focusing on the feasibility of such a technique than suggesting its use.

REPLY. Toxicity is always a question of context. Any element or chemical compound is toxic in the wrong circumstances. Chlorine is very common in the natural environment without being toxic, the main concern is anthropogenic organochlorine compounds. We have done modeling which demonstrates this is not an issue. Chlorine can be and is handled safelty in laboratories around the world.

The wording of the manuscript should be strongly improved by careful proofreading! In addition the technical quality is not yet good enough, spaces are missing, brackets are not closed, superscripts are not used, etc., please carefully check for this.

REPLY. Thank you, the manuscript has been carefully proofread, also by a native speaker, and corrected as needed. The corrections can be seen in the track changes/latexdiff version.

In addition the structure of the experimental section should be streamlined to improve readability (see

below).

REPLY. Thank you, we have done as suggested.

Further, I have a number of specific comments and technical corrections the authors should consider before publication is feasible.

*Specific comments:*

The abbreviations applied for setups, experiments and flasks in Table 1 +2 are confusing. Why not using one simplified setup (Figure 1), which guides the reader in a better way (see specific comments below).

REPLY. Done.

The experiments in the results section should be streamlined, only experiments required to tell the main story should be selected, ordered in a well-motivated manner, and main results provided in the main text of the manuscript. Currently all results are mentioned in the results section but removal efficiencies only given in the appendix (D1 – D3).

REPLY. We have extracted the relevant experiments from table D1-D3 and now present them in the main text in a summary Table 3 (Exp C, E-I.))

The modelling and experimental results should be combined to cross-validate each other. Why not displaying the modelled functions in e.g. Figure 4?

REPLY. The modelling and experiments are cross-validated in figure 6. The text now makes this more clear.

*Technical corrections:*

Page 1 Line 19: Possibly "High-resolution instruments targeting at specific rovibrational transitions …"

or similar.

REPLY. Thank you, we have written 'High-resolution instruments based on specific rovibrational transitions are becoming available to characterize the abundance of rare isotopocules within gases.'

Page 2 Line 27: Please change to $\delta^{15}N^{\alpha}, \delta^{15}N^{\beta}$ here and elsewhere in the text.

REPLY. We have changed this example and throughout the text.

Page 2 Lines 27 – 28: The term "biological activity in agricultural soils" is very unspecific; in addition the message of the sentence is unclear. Would the following sentence fit better? "These instruments are often used to measure isotopic signatures of $N_2O$ emitted from soils (Ibraim et al. (2019a), Wolf et al. (2015)), which can help to differentiate different microbial and abiotic production pathways."

REPLY. Thanks very good, we have changed as suggested. ' These instruments are often used to measure isotopic signatures of N2O emitted froms oils, (Ibraim et al. (2019a), Wolf et al. (2015), Yu et al. (2020)), which can help to differentiate distinct microbial and abiotic production pathways. '

Page 2 Lines 29 – 30: The subsequent sentence might be changed to: "$N_2O$ formation in soils is commonly accompanied by production / uptake of other trace gases such as $CH_4$, $CO_2$, and water vapor (Erler et al. (2019), Ibraim et al. (2019b))."

REPLY. Thanks we agree and have changed as suggested.

Page 2 Line 30 – 31: The message of the sentence "In some samples, changes in $CH_4$ and $CO_2$ can exceed 1.8 ppm and 200 ppm, respectively (M. Zimnoch and Rozanski (2010))." is unclear and the numbers seem arbitrary. Please provide a more profound statement on $N_2O$, $CH_4$, $CO_2$ concentrations above different soils. I would suggest to give a high and low emission scenario for both $CH_4$ and $CO_2$. This should give the reader a feeling on usual $CH_4$ and $CO_2$ changes along with targeted $N_2O$ concentration (and isotope) changes.

REPLY. Thank you, we have modified to present the argument more clearly. 'These variations complicate measurements. An example of the relevant variation of CO2 and CH4 can be found in the work of (M. Zimnoch and Rozanski (2010)) where the background level of CH4 and CO2 at 1.8 ppm and 380 ppm, can change suddenly to levels above 3.6 ppm and 560 ppm.'

Page 2 Line 31 – 33: A number "5 ‰" is provided without a statement on whether the main interference is from $CH_4$ or $CO_2$ and which delta values are affected most. Please give the deviations for a specific delta value for the high and low $CH_4, CO_2$ emission scenario (statement above).

REPLY. We have added an explanation for 5 ‰ as part of our modification.

Page 2 Line 33: The statement "improve the accuracy by controlling these interferences" is questionable? Possibly "Improve the accuracy by either removing the interfering trace gases (i.e. $CH_4, CO_2$) or the target substance ($N_2O$) or the analysis of interferants and implementation of a correction algorithm."

REPLY. We have changed the wording to give the correct description which is 'accounting for' rather than 'controlling'. We have also added a descrition of trapping N₂O and releasing it into a well-defined matrix.

Page 2 Line 35: How should the "a careful determination of the calibration curve.(Kantnerová et al. (2020))" help? Please rephrase or delete this sub-sentence.

REPLY. Deleted.

Page 2 Line 38: Place replace "continued" by "on-line" or "monitoring applications" or similar.

REPLY. Changed to 'on-line'

Page 3 Table 1 and 2: The wording of setups should be simplified to support the readers, e.g. using "Xe lamp" instead of "XPXL", or "Photochemical Device" instead of "STH-MFC-PD", etc.? The different setups are quite similar, would it be possible to show one setup only and mention difference in the others?

REPLY: We have changed the setup illustration to be one that is exemplary for all the setups and added the difference in the figure text. The names for the setups have been changed to setup 1-4 and are summarized in Table 1.

The experiments are labelled (A-I) same than the flasks (A-D), please change one label.

REPLY: We have changed the gas flask numbers to roman numbers

Page 3 Line 64: The end of the sentence "… applying the method to measurements of $N_2O$." should be reformulated.

REPLY: Has been changed to: "With the method developed and refined, a final set of experiments is conducted using a Picarro CRDS model G5131-i, capable of measuring N2O mixing ratio and its isotopic abundance"

Page 3 Line 65 – 66: The last sentence of the introduction should be rephrased, e.g. "the measured isotopic data of isotopes", "compared to the stable values", etc.

REPLY: We have rephased it to: "The measured values of , $\delta^{15}N^\alpha, and\ \delta^{18}O$, subject to methane interference, are compared to data corrected for methane levels, as these corrected isotopologue levels remained stable across the experiment")

Page 3 Line 67: The structure of the method section should be rethought: 2.1.1 Methane experiments; 2.1.2 Post photolysis scrubbing. I assume Post photolysis scrubbing is part of the Methane removal experiments, so should be integrated in 2.1.1?

REPLY: "Post photolysis scrubbing" is a part of "methane experiments# and it is set as a subsection to that.

Page 3 Line 71: What is a "Flow-Controlled Chlorine Waste", please rephrase.

REPLY: The Chlorine waste line is either flow controlled or pressure controlled. It is made more clear in Table 1.

Page 3 Line 70 – 73: The section should be rephrased considering the final.

REPLY: We have included a table (table 1) providing an overview of the setups and the experiments. With this we have also rephased to "Four different variations of the setup seen in Figure 1 are used during our experiments, summarized in Table 1 together withwhich experiments they were used for."

Page 4 Figure 1: Please show one exemplary setup and mention differences for the others?

REPLY: Figure 1 have been edited to a general phtochamber and differences are described in the figure text.

Page 4 Line 74 – 82: First there are some sentences on gas cylinders flows and setup, thereafter a section "Manifold" dealing specifically with the setup? Please combine and rephrase both sections.

REPLY: We have delete the " manifold" headline and combined

Page 5 Line 98: The statement "they were found to likewise remove $HCl$ $and$ $Cl_2$" Is unspecific for such an important question. How was this determined and what happens with the Picarro analyzer if it is flushed with HCl or $Cl_2$?

REPLY: The Picarro can not handle corrosive materials. The flow after the trap is tested for $Cl_2$ with the $Cl_2$ monitor. None was observed. Additional samples were also collected for measurements with GC-MS, and here the introduction of the ascarite trap was shown to remove HCl as well.

Which chlorinated species pass the Ascarite trap and are scrubbed with the activated carbon trap, please specify? Is it advisable to purge these species through a Picarro CRDS analyzer?

REPLY: The flow is tested with the GCMS before and after the ACT and the Ascarite. It is shown that $CCl_4$, other Chlorinated VOCs, and HCl are removed.

Page 5 Section 2.1.3: The authors should mention how measurements were referenced to scales? Were the delta values of the applied $N_2O$ gases known and deviations analyzed for the $CH_4$ addition experiments and the $CH_4$ removal experiments?

REPLY: Unfortunately no, as the calibration gasses we had intended to use for scaling the delta values had not arrived when we began our experiments. So the size of the delta values is indeed unbound. Fortunately, this is not an issue as what we present is the relative change. We have normalize our delta values to methane corrected delta values.

Page 7 section 3.1: The author show results of experiment H first, but do not indicate how this is motivated?
REPLY: Exp. H is a combination of change residence time and increased power input. We have included this statement as explanaton for our choice: "As an example of our data, we present the results formexperiment H, Figure 3, during which we achieved our highest level of removal."

In addition, main results (Table D1 – D3) are hidden in the appendix. I would suggest to streamline the structure of the results section, and provide important results in the main text in a wellmotivated order.

REPLY: We agree, and we made a summary table of this in the main text. See Table

Page 7 Line 151: "experiment 3 step 5" is this "experiment C step 5"?

REPLY: We agree, it has been changed to C.

Page 8 Figure 3: The authors mention in the legend that "the experimental step is indicated at the top" but the experiments are named Sx, while in the text the terms Hx are used, please change?

REPLY: We have changed all experimental figures to be named Ax, Bx, etc

Page 8 Line 160 - 162: The sentence should be reformulated, was it really a pressure drop in the cylinder?
REPLY: We are indeed wrong to say the cylinder, as the pressure drop occurred at the regulator on the cylinder, and we have changed it accordingly.

If a pressure reduction valve was used this seems unplausible? As a $Cl_2$ sensor was used this should have been noted during experiments

REPLY: While we agree that a pressure reduction valve should not lead to this, unfortunately, we failed in applying best practice when installing the regulator, as it had not been sufficiently flushed through with dry air before the experiment. It is therefore assumed that a small amount of humidity was present on the inside, which should have been avoided at all cost when working with $Cl_2$, as it allows for the initiation of corrosion. The pressure drop from the regulator is, therefore, explained as corrosion affecting the regulator. While this is a strong point against the use of $Cl_2$, we will argue that it can be done properly if the equipment is correctly prepared. And explanation of this has been included in the manuscript.

Page 10 section 3.1.1: Figure 5 provides the main finding of the manuscript with respect to consistency of delta values for $CH_4$ addition / correction and $CH_4$ removal / correction.

*I have some concerns:*

1.The authors did not achieve complete $CH_4$ removal, which would be needed to waive the need for $CH_4$ interference correction. What was the reason for this as experimental details for quantitative removal are known from the preceding section?

REPLY: The experimental conditions during the $N_2O$ experiments did not allow for the same removal conditions as used when 99% removal was achieved We have included an explanantion of this: "In the $N_2O$ experiments it was not possible to apply the same conditions that lead to the highest levels of removal presented in the earlierexperiments. The reason for this was the addition of the G5131-i increased the minimum flow through the photo-chemical-device, thus decreasing the maximum residence time. Additionally not having a high concentration $N_2O$ source capped the dilution, as the $N_2O$ needed to remain in the linear range of the G5131-i. The limit on the dilution therefore also limited the concentration of $Cl_2$ available. With a higher concentration $Cl_2$ source available and a properly prepared regulator, the setupwould have been able to deliver sufficient $CH_4$ removal for more than 24 hours, at which point the ascarite trap would needreplenishment"

2) The indicated $\delta^{18}O$ and $\delta^{15}N$ numbers are highlight unplausible, not normalized to scales? This might be ok for a feasibility study, but why are results not normalized to the $N_2O$ isotopic composition without $CH_4$ addition?

REPLY: Certainly correct, and as stated in response to one of the previous comments, we have normalized it to the $CH_4$ corrected values.

3) Figure 5 shows mainly instrumental noise of the CRDS analyzer, with some plausible drops in the signal, when $CH_4$ was removed.

REPLY: Indeed the value of $\delta^{18}O$ seems dominated by instrumental noise. We have inserted the table (as Table 4) of the experimental results in the maintext to better convey the improvement.

4) Why did the authors observe variations in $N_2O$ concentrations? This a very critical observation if $N_2O$ was removed by a chemical process and would question the approach. Please comment on this?

REPLY: Variation in $N_2O$ was indeed observed, but we are confident it can be explained by the drop in $Cl_2$ flow. Firstly because the observed change was an increase in concentration and therefore does not point to a removal. Secondly, because similar increases were observed in the methane concentration at the same time. Variation also took place during dark periods indicating it is not due to chemical reactions.

table from the appendix to the main text.

---

## Author Comment (AC2)

Comments on "Photochemical method for removing methane interference for improved gas analysis" by Merve Polat et al.

General comments:

This manuscript describes a photochemical scrubbing method, using $Cl_2$, for the removal of $CH_4$ in whole air samples. This described method is intended to enable accurate determination of the isotopic composition of $N_2O$ ($\delta^{15}N^\alpha$, $\delta^{15}N^\beta$, and $\delta^{18}O$) by minimizing the spectroscopic interference arising from $CH_4$ (among several others), which poses immense challenge. This method indeed shows potential as the removal of $CH_4$ through this photochemical scrubbing does not alter the matrix composition dramatically. In addition to the experimental work, the authors have also complemented their experimental results with predictions using a kinetic model studying all the dependencies. This method in general can be used for any application requiring removal of $CH_4$ (and other hydrocarbons/VOCs), and is not just limited to the measurements involving $N_2O$. And hence is an important development that the scientific community could benefit from.

Although the content of the manuscript is very interesting, consisting of appropriate method-development related experiments and complementing model prediction, the manuscript itself is difficult to follow at times due to: the use of too many abbreviations, and having to go back-and-forth between the main section of the manuscript and the supplementary section where part of the information is.

REPLY: We agree that the current structure of the manuscript forces readers to go back and forth between the supplementary section and the main text, and we have corrected for this issue. With regards to the use of abbreviations, we have attempted to limit our use of them.

Additionally, there are sections where the texts require rephrasing to make the content more understandable. Please see details in the specific comments section.

REPLY: We are happy to apply your suggestions to make the text more easily readable

While this proof-of-concept method is aimed towards reducing the interferences of $CH_4$ during the measurements of the singly-substituted isotopologues of $N_2O$, the experiments and results shown to demonstrate the applicability is very limited. Since the removal efficiency of $CH_4$ achieved is never ~100%, discussion on its implication was not evaluated thoroughly.

REPLY: It is unfortunate that the parameters available to us when doing the measurements of $N_2O$, did not allow for the removal of >98% we achieved during the parameters testing. Regardless of this, we now agree that the acquired results should be discussed in more detail and be related clearly to our declared goal. So we want the variation in methane reaching the Picarro G5131-i to be less than 0.2 ppm as the size of this variation will then be below the detection limit of the instrument.

This is particularly concerning because one has to then always co-measure methane, which is partially scrubbed, in order to incorporate any possible $CH_4$-dependent correction. So does the method provide any benefit over performing a careful $CH_4$-dependent interference correction?

REPLY: With the presented removal during the $N_2O$ experiment, indeed, it would still necessitate co-measurements of $CH_4$. But we will argue that by improving upon the photochamber, it would be possible to continuously remove more than 98% of $CH_4$. We have added the arguments for this in the manuscript, in combination with the explanation for why we did not achieve >98% removal as seen in the earlier experiments. (see reply to first concern from Reviewer one)

Additional experiments showing the repeatability expected from this method using isotopically-calibrated $N_2O$ samples was also missing.

REPLY: Fair point, that we do not have data to describe.

The manuscript in its current form requires considerable rework and I would recommend publication after all concerns have been addressed.

*Specific comments:*

Line 74 and Table 2: How is tank A calibrated for $[Cl_2]$? Is this a commercial product?

REPLY: Yes, it is a commercial product.

Line 82: The chlorine detector is rated for 0-20 ppm, so how were chlorine concentrations determined in experiments done above 20 ppm, shown later in the manuscript?

REPLY: The Cl concentration is theoretically determined

Line 85: LED already stands for Light emitting diodes, so should be mentioned "LED" only and not "LED diodes". Please correct this throughout the manuscript.

REPLY: We have changed this throughout the manuscipt.

Line 93: Please provide supplier details/ part number for Krytox™.

REPLY: We have added (DuPont GPL 205 Krytox Performance Grease )

Line 96: Magnesium perchlorate is $Mg(ClO_4)_2$

REPLY: We agree, we have changed it.

Line 100: Please provide supplier details, part number, specification for the activated carbon trap used.

REPLY: We have added (Bead-Shaped Activated Carbon, KUREHA Corporation)

Line 103: "A final set of experiments is conducted using a Picarro CRDS model G5131-i, capable of measuring $N_2O$ mixing ratio and its isotopic abundance."

REPLY: We have changed it to the line as suggested "A final set of experiments is conducted using a Picarro CRDS model G5131-i, capable of measuring N2O mixing ratio and isotopic abundance"

Lines 150-152: Throughout the manuscript, the steps are referred to as S1, S2.... and not as in your example C5 (line 152), please check and revise accordingly to be consistent. Line 155: "H5": please see my previous comment.

REPLY: We have edited the steps to be referred to as their experimental name.

Lines 159-162: This paragraph somehow feels very unlinked with the previous paragraph. Please explain the "issue" by pointing the reader to the graph, what exactly to look at? How does the build-up of $H_2O$ happen? Why is the $Cl_2$ raw data not shown along with?

REPLY: We have added the call to attention "…as can be seen from the slope in step H1." To the explanation. The build-up of moisture arose from leaving most of the setup system open over night, which allowed lab air to enter. With regards to the lack of raw $Cl_2$ data, the sensors used for $Cl_2$ were a "cheap" monitor intended for safety and did not come with logging. This is also the reason it, unfortunately, was not able to measure $Cl_2$ concentration as high as we would have liked. Our $Cl_2$ measurements were therefore used to confirm that the concentration of $Cl_2$ had not changed compared to the onset. Unfortunately, as we described in the paragraph in question, we did observe a drop in $Cl_2$, but we, unfortunately, do not have the $Cl_2$ data to show for it..

Figure 4: Typically, when you have units shown on the axis label, you don't have to show them on the tick labels, so the % signs on the ticks can be removed. And the abbreviation RE has not been introduced anywhere in the text, so please include this.

REPLY: We have changed the figure axis as suggested and introduced RE in the figure text.

Line 187: What is d here? Please define your notation. Is it delta that you are referring to? If so, please describe how often you measure your reference/ calibration etc. Please check and change this throughout the manuscript.

REPLY: That is indeed intended to be delta. It will be corrected throughout the article. For explanaiton we have written; "The delta values are self-referenced to the gas without the addition of CH4."´As for the measurements of reference/calibration gas, we certainly desired to do it, to follow the recommendations by the paper of Harris. Unfortunately, our calibration gasses first arrived after the completion of these experiments. We argue that while calibration of the isotopologues would be preferable, it is in the end unnecessary as we are only determining the relative change of the acquired delta values.

Line 187-188: "The results are from experiment L, where a sofnocat trap had been was installed to remove the CO formed by the $CH_4$ oxidation."

REPLY: We have changed it to "The results are from experiment L, where a sofnocat trap had been installed to remove the CO formed by the CH4 oxidation"

Line 189: "…it was found that the isotopic enrichments …" Please introduce this to the reader why they should expect isotopic enrichment and not depletion in δ15Nα and δ18O.

REPLY :We see the error we invite by our phrasing, as we are not talking about an enrichment but rather the actual isotopologue level. We have rephrased to "it was found that the isotopologue levels remained stable through the oxidation (grey line)."

Line 190: How stable is the oxidation process in a prolonged time period, e.g. during a continuous 10 hr measurement period, and in practice you would really turn it ON all the time during a measurement. How much of the variability in $[Cl_2]$ translates into your final measurement uncertainty?

REPLY: The ideal  is to turn it on and leave it running, but we have not left it running for multiple hours. The longest continuous running was for roughly 100 minutes (as shown in fig 3), where the performance was stable. We expect that this performance could be maintained for several hours but we have not done the experiment to show it. This does touch on another problem that would arise from long-continued measurement, which is the Ascarite trap will be used up and in need of a replacement after roughly 22.5L had passed it. (corresponding roughly to one day's worth of measurements at 15ml/min). In our description of the N2O results we have written "With a higher concentration Cl2source available and a properly prepared regulator, the setupwould have been able to deliver sufficient CH4removal for more than 24 hours, at which point the ascarite trap would needreplenishment."

And for the final subquestion, how does the variability in $Cl_2$ affect the overall uncertainty? While it will depend on the concentration and the amount of $Cl_2$ introduced, in our experiment we have estimated the uncertainty of the $Cl_2$ flow to be 8.3%, which is the greatest source of uncertainty in our work. The uncertainty of the concentration of $Cl_2$ comes to be around 10%.

The solution to this would be to make a better and more stable supply of $Cl_2$, and/or have a high enough concentration source so that the variation in $Cl_2$will not result in any relevant variation in methane removal potential.

Line 193-194: How is the variation in $[N_2O]$ due to variation in $[Cl_2]$, is it not due to dilution? And correct the spelling of variation in Line 194.

REPLY: Correct that would be a more accurate description, as the variation in $Cl_2$ is also due to changes in the dilution. We have rephrased to avoid confusion as "Variations of roughly 5% were observed in [N2O] but are accounted for by variations in the flow of [Cl2], thus changing the dilution, rather thanformation of N2O due to the photochemistry."

Section 3.1.1 (in general): Why was the method with the highest removal efficiency not used here?

REPLY: We have answer the same question from the first reviewer, and included an explanation in the text.

Line 229: "… that an increase in $Cl_2$ concentrations increases the $[CCl_4]$ production (see Figures 7a, 7b and 7d.)"

REPLY: Yes, we have changed as suggested to "Figure 7a showsthat an increase in Cl2concentrations increases the [CCl4] production."

Figure 7 (caption): "The Removal efficiency of methane depletion (Black), …" should be "The removal efficiency of methane (Black), …"

REPLY: Yes, we have changed it as suggested to "The removal efficiency of methane (Black), [\ce{CCl4}] (Red) and [Cl] (Grey) is shown in the Figures a-d."

Line 233-234: If you use NaOH to remove $CO_2$ from a sample, the matrix changes significantly. To what level of matrix alteration not a problem?

REPLY: The removal of $CO_2$ has an effect, but as $CO_2$ is only (in the high emission situation) 560ppm, removing it results in a 0.056% increase in the concentration of the remaining gasses, which can be worth considering, though we do not agree with it being significant. $H_2O$ removal however does pose a significant alteration to the matrix, and we should include a recommendation of measurement of $H_2O$, so this matrix alteration can be accounted for.

Lines 234-235: "The NOx concentration in our experiments is insignificant and hence these reactions have not been included in the model."

REPLY: Yes, we have changed it as suggested to "The NOx concentration in our experiments is insignificant and hence these reactions have not been included in the model."

APPENDIX

Line 252: What is the typical concentration range of $Cl_2$ produced by this method? Please elaborate this and describe the calibration and monitoring/ data recording method for $Cl_2$.

REPLY: We can expand further on this proof-of-concept experiment, but the assessment of the $Cl_2$ production is made difficult by high variation and an upper limit of detection for the $Cl_2$ monitor at 20ppm. We have elaborated by writing: "The ambient air standard was enriched in Cl2by in-situ production of Cl2, ranging from 1ppmto <20ppm, through electrolysisof a saltwater mixture."

Line 285-286: Please rephrase.

REPLY: We have rephased to"28 LED (385 nm), UV LED LAMP-VAOL-5EUV8T4, diodes was installed in the chamber, directed at a quartz tube, OD: 4mm L: 20cm, placed through the chamber."

Lines 331-336: Please avoid repeating texts already used in the main body of the manuscript (lines 89-93). REPLY: Well spotted. We will corrected for this.

Figure B1: Abbreviations are typically introduced once, the first time they come up in the document. So please don't expand your abbreviations every time you describe a figure.

REPLY: Noted.We have simplified the figures and removed the multiple sets of abbreviations.

Line 354: How does CO interfere with $N_2O$, please elaborate this and remind the reader which isotopologues are specifically affected.

REPLY: Certainly, we have included the following explanaiton "The trap was installed to prevent effects on the N2O isotope signal from CO, aspresented in (Harris et al. (2020)) the presence of CO 1 ppm gives rise to an erroneous offset in the observed isotopologue values of 1.2, 2.4 and 0.4 ‰ for δ15Nα, δ15Nβ, and δ18O respectively."

Figure C1: Why repeat a figure when you can refer to Figure 2?

REPLY:We agree, and this repeat has been removed.

Lines 407-410: Please rephrase this paragraph and elaborate on "This effect…". The explanation is not clear.

REPLY: Our attempts to calculate the photolysis rate based on the parameters, did not scale linearly with power input. This offset is what we refer to as the "effect", and upon rereading we agree that the section is not clear, especially as the function presented is linear. We have rephrased it to "Where W is the power supplied to the diodes, and values for the constants a and b are fitted in the model to match the experiment. The function C10 accounts for additional variations such as effects due to temperature, the cross-section area ofthe quartz tube, the conductance of the photochamber and the quality of the distribution fit. This is reflected in the constants a and b varying in response to changes in these parameters. As this is used a as simple empirical stand-in function we do not intend to speculate further on how these changes change the constants."

Table D4 (Caption): "… refeer to the three isotypes of $N_2O$." should be something like "… in ‰ refers to the three isotopologue measurements of $N_2O$."

REPLY: We have adopted the suggested change and moved the

---

## Author Response (AR2)

28 September 2021

Dear Editorial Office,

Thank you very much for the opportunity to publish our manuscript with AMT, it is greatly appreciated.

We have updated the nomenclature as you've asked and also corrected the information in the references. To the best of our knowledge everything is now complete.

With best regards and on behalf of the coauthors,
Matthew S. Johnson